# Distinct information conveyed to the olfactory bulb by feedforward input from the nose and feedback from the cortex

Joseph D. Zak [1,2] ✉, Gautam Reddy [3,4,5], Vaibhav Konanur [1] & Venkatesh N. Murthy [5,6,7]

Sensory systems are organized hierarchically, but feedback projections frequently disrupt this order. In the olfactory bulb (OB), cortical feedback projections numerically match sensory inputs. To unravel information carried by these two streams, we imaged the activity of olfactory sensory neurons (OSNs) and cortical axons in the mouse OB using calcium indicators, multiphoton microscopy, and diverse olfactory stimuli. Here, we show that odorant mixtures of increasing complexity evoke progressively denser OSN activity, yet cortical feedback activity is of similar sparsity for all stimuli. Also, representations of complex mixtures are similar in OSNs but are decorrelated in cortical axons. While OSN responses to increasing odorant concentrations exhibit a sigmoidal relationship, cortical axonal responses are complex and nonmonotonic, which can be explained by a model with activity-dependent feedback inhibition in the cortex. Our study indicates that early-stage olfactory circuits have access to local feedforward signals and global, efficiently formatted information about odor scenes through cortical feedback.

Volatile odorants are sensed by olfactory sensory neurons (OSNs) in the main olfactory epithelium of mammals[1]. Each OSN expresses only one odorant receptor (OR) type out of a large ensemble but can sense many ligands with different sensitivities[2,3]. The large number of receptor types with broad selectivity is thought to underlie the combinatorial capacity of the olfactory system to sense a substantial number of odors in the natural world. Axons of OSNs expressing the same OR converge on glomeruli in the olfactory bulb (OB)[4–6]. This convergence is likely to help in signal averaging, and postsynaptic projection neurons called mitral/tufted (M/T) cells receive focused excitatory inputs from single glomeruli[7,8]. A complex network in the OB, which includes many types of inhibitory interneurons[9], transforms the incoming odorant information before it is sent to downstream brain regions, including the piriform cortex (PC). An intriguing feature of the early olfactory system is the dense axonal feedback projections from olfactory cortical areas to the OB, which brings processed cortical information back to the earlier stages[10–13]. It is not clear how the feedforward information from the nose and feedback from the cortex interact in the OB.

The elaborate and often hierarchical organization of sensory systems is widely thought to help achieve efficient coding of information[14–16]. One way in which brains are thought to achieve efficient coding is by making responses of a neural population uncorrelated and of similar sparsity for a wide range of stimuli[15–20]. In the olfactory system, the repertoire of ORs is fixed in the genome and their responses to odorant stimuli may be inefficient - for example, certain odorants may activate many receptors and others may activate very few[21–24]. To make the representation more efficient, circuits in different

[1]Department of Biological Sciences, University of Illinois Chicago, Chicago, IL 60607, USA. [2]Department of Psychology, University of Illinois Chicago, Chicago, IL 60607, USA. [3]Physics & Informatics Laboratories, NTT Research, Inc., Sunnyvale, CA 94085, USA. [4]Department of Physics, Princeton University, Princeton, NJ 08540, USA. [5]Center for Brain Science, Harvard University, Cambridge, MA 02138, USA. [6]Department of Molecular and Cellular Biology, Harvard University, Cambridge, MA 02138, USA. [7]Kempner Institute for the Study of Natural and Artificial Intelligence, Harvard University, Allston 02134, USA. ✉e-mail: jdzak@uic.edu

brain regions, including the OB and the PC may perform computations such as normalization or whitening[25–28]. While circuits in the OB are likely to achieve some of the computational goals[29], the PC is better situated for more global associations.

Principal cells in the PC integrate information from multiple glomerular channels in the OB, conveyed by M/T cells[30–32]. This information is then reformatted through recurrent excitatory and inhibitory circuitry in the PC, presumably to aid odor perception. For example, the fraction of neurons in the PC responding to different monomolecular odorants is relatively constant[25,33–35], even though these odorants could activate very different densities of OSNs. Similarly, different concentrations of a given odorant activate a similar fraction of PC neurons[25,35–37]. Correlations in the representation of monomolecular odorants are also restructured in the PC[26].

While previous studies point to normalization of olfactory responses in the PC[38], several key features remain unknown. For instance, even though the olfactory environment consists of a complex mixture of chemicals, we currently lack an understanding of how cortical neurons represent realistic odorant mixtures, which elicit complex interactions even in the OSNs[24,39–43]. Importantly, in the context of feedback, it is unclear which of the different computations ascribed to the PC, related to odorant identity[25,44], quality[45], attention[46], and predictive coding[47], are conveyed back to the OB.

In this study, we make use of a diverse set of olfactory stimuli to ask how their neural representation is transformed from the OSNs to feedback from the PC to the OB. We imaged the activity of mouse OSNs in the olfactory epithelium in response to pure odorants, as well as a wide variety of mixtures of odorants containing up to 12 components. We also imaged OSN responses to variations in odorant concentration over 4 orders of magnitude. We measured how expanding sensory input influences the activity of PC feedback by imaging its axonal projections to the OB. Our results reveal that cortical feedback axons bring back strongly normalized and decorrelated information about diverse odorant mixtures and concentrations to the OB, which can be combined with feedforward signals to influence M/T cell responses.

## Results

### Odorant tuning profiles of cortical projections to the OB

Principal neurons in the PC integrate inputs from multiple glomeruli (Fig. 1A) and can respond to odorant stimulation with either an increase or decrease in activity relative to their baseline activity[12,25,37]. We began our study by systematically measuring the responses of cortical feedback axons in the OB of awake mice to a panel of monomolecular odorants.

A cocktail of two viruses was injected into the anterior portion of the PC (see Methods) to drive the expression of the calcium indicator GCaMP6f (Fig. 1B) and the fluorescent marker tdTomato (see Supplementary Fig. 1), which was used to identify regions containing infected projections and for motion artifacts compensation. A *post hoc* analysis revealed dense indicator expression in the somata of neurons in layer 2/3 of the PC (Fig. 1C), as well as their axonal projections in the granule cell layer of the OB (Fig. 1D). Projections could also be observed to reach the glomerular layer of the OB, although at lower densities.

In living mice imaged through cranial windows, we observed dense indicator expression at the interface between the external plexiform and granule cell layers (Fig. 1E). To measure functional responses, we selected a standardized odorant panel that has previously been used in our laboratory for both behavioral and physiological studies[24,48] (Supplementary Table 1). Odorants were delivered for two seconds each in a pseudorandom order with intertrial intervals of at least 20 seconds between odorant presentations. The odorant-evoked responses were measured in individual axonal boutons by generating ROI masks from spatiotemporal correlograms (refs. 49,50; Fig. 1F). Within individual imaging fields, enhanced and suppressed bouton responses were spatially distributed throughout the area

imaged and their response kinetics varied by odorant identity (Fig. 1F–H).

In our characterization of odorant tuning profiles of cortical feedback, we collected data from 832 boutons in nine imaging fields from six mice. Exemplar odorant response characteristics are shown in Fig. 1. For all bouton-odorant pairs that were significantly stimulus-modulated, response polarities were typically conserved over the odorant panel (Fig. 2A). Individual bouton responses were generally either enhanced or suppressed across the odorant panel, and a smaller fraction showed mixed responses (37.5% enhanced, 38.0% suppressed, 24.5% mixed). These proportions are significantly different from chance, which would be $6.5 \pm 0.6\%$ enhanced, $11.7 \pm 0.3\%$ suppressed and $81.3 \pm 1.0\%$ mixed, if odor-bouton responses are independently drawn from the distribution shown in Fig. 2B (see Methods; deviations are 99% confidence intervals). Our findings are consistent with other studies that report that response polarity is conserved between stimuli within the PC and descending axons terminating within the OB[12,25,37].

Most boutons did not respond to any particular stimulus ($72.2 \pm 1.3\%$ unresponsive, $n = 16$ odorants; Fig. 2B). However, of the boutons that were odorant-modulated, suppressed responses were more frequent than enhanced responses ($15.4 \pm 0.9\%$ suppressed vs. $12.4 \pm 0.9\%$ enhanced, $n = 16$ odorants, $P = 0.038$, sign-rank test; Fig. 2B). We next considered the tuning widths of individual boutons by estimating the number of effective odorants, that is, those that generated a response significantly above or below baseline activity (see Methods). Boutons that were suppressed by odorant stimulation were more broadly tuned than those that were enhanced ($3.9 \pm 0.2$ effective odorants for suppressed boutons and $3.2 \pm 0.2$ effective odorants for enhanced boutons; $n = 16$ odorants; $P = 0.009$; rank-sum test; Fig. 2C).

Suppressed boutons responded more strongly to odorant stimulation than enhanced boutons ($0.67 \pm 0.01$ $z$-score for enhanced boutons, $n = 1652$ bouton-odorant pairs; $0.71 \pm 0.01$ $z$-score for suppressed boutons, $n = 2046$ bouton-odorant pairs; $P < 0.001$; Kolmogorov–Smirnov test; Fig. 2D); however, when considering each odorant, the average population responses of enhanced boutons were similar to suppressed boutons ($P = 0.39$; Kruskal–Wallace test; Fig. 2E). In both suppressed and enhanced boutons, there was a significant positive relationship between the fraction of activated elements and their mean activity ($r = 0.81$ for suppressed boutons, $P < 0.001$; $r = 0.57$ for enhanced boutons, $P = 0.021$; Fig. 2F). Overall, in response to a monomolecular odorant, cortical feedback axons exhibited a balanced profile of activation and suppression that was statistically similar for all stimuli.

### Cortical feedback boutons are more broadly tuned than individual feedforward input to the OB

To estimate the relative sparsity of bouton responses, we compared their properties to those of OSNs using the same odorant panel. In OMP-GCaMP3 mice, we used a bone-thinning procedure to gain optical access to individual OSNs within the olfactory epithelium (Fig. 3A; refs. 24,40,51). A schematic of the relative window locations can be found in Supplemental Fig. 2. We then delivered the same 16 odorants as we did for experiments imaging feedback projections to the OB.

Cortical neurons are expected to be more broadly tuned than sensory cells given the convergent circuit architecture of bulbar inputs to the PC[30,52,53]. Our data, which makes use of the same odorants to stimulate both OSNs and cortical projections to the OB, indeed supports this expectation. We found that OSNs responded to $3.15 \pm 0.15$ odorants in our panel, while feedback projections responded to $4.45 \pm 0.13$ odorants ($n = 377$ OSNs, 832 boutons; $P < 0.001$; rank-sum test)). However, somewhat unexpectedly, cortical projections are, on average, sensitive to only ~1.5 more odorants out of 16, far less than the estimated convergence of ~200 glomeruli per cortical neuron[53,54].

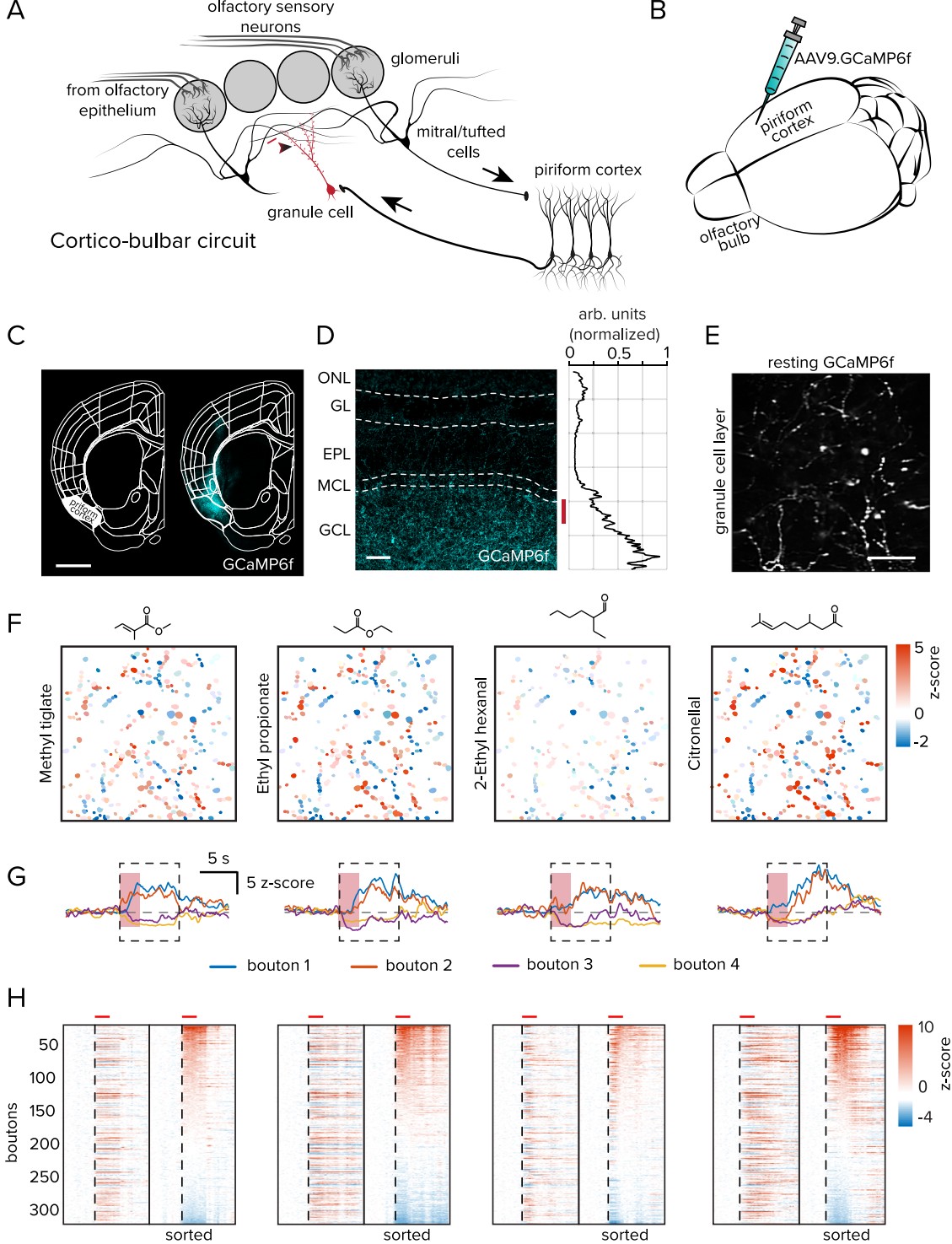

**Fig. 1 | Viral expression of fluorescence indicators in the anterior PC. A** Circuit schematic of cortico-bulbar connectivity. **B** AAV9.GCaMP6f was injected bilaterally at two sites in the anterior region of the PC. **C** Representative image of a coronal section of a mouse used in subsequent experiments GCaMP6f expression is targeted to the PC, scale bar = 1000 μm. **D** Representative image of GCaMP6f expression in cortical projections to the OB. The red bar denotes the typical imaging depth for in vivo experiments, scale bar = 100 μm. Right, the normalized fluorescence intensity in each layer of the OB. Olfactory nerve layer (ONL), glomerular layer (GL), external plexiform layer (EPL), mitral cell layer (MCL), and granule cell layer (GCL). **E** Left, Representative image of image of GCaMP6f resting fluorescence of a typical imaging field, scale bar = 20 μm. **F** Responses to four

selected odorants mapped onto ROI segments. **G** Temporal modulation of GCaMP6f responses in four selected ROIs from each of the four odorants in part **F**. Response polarity is conserved across different stimuli. The red vertical shaded area denotes odorant delivery time. The dashed box indicates the response averaging window for subsequent analyses. **H** Responses of each of the 341 boutons in the imaging field above to the same four odorants. Left, boutons are sorted by spatial location in the imaging field, right, traces are sorted by their mean response amplitude. The vertical dashed line denotes the odorant onset and the horizontal red bars above are the odorant duration. The exemplar images are available in the data repository noted in the Data Availability Statement.

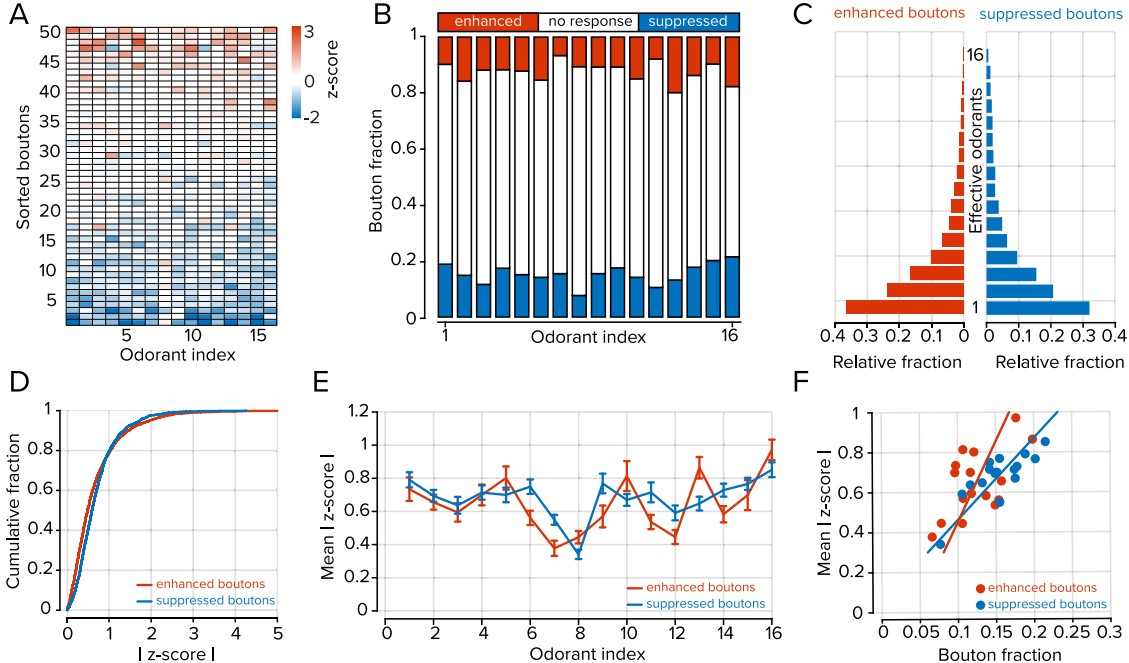

**Fig. 2 | Odorant tuning properties of cortical projections to the OB. A** Odorant tuning profile of 50 randomly selected boutons selected from all imaging fields. Boutons are sorted by their mean response amplitude across all odorants. **B** Fraction of responding boutons for each odorant. **C** Distributions of effective odorants for boutons that showed net enhanced (3.2 ± 0.1 odorants) or suppressed (3.9 ± 0.2 odorants) responses ($n = 16$ odorants; $P = 0.009$; rank-sum test). **D** Cumulative distributions of all stimulus-modulated responses at all odorants for enhanced and suppressed boutons ($P < 0.001$; Kolmogorov–Smirnov test). **E** Mean response of all significantly modulated boutons separated by response polarity for each odorant ($P = 0.39$; Kruskal–Wallace test), error bars represent standard error of the mean (s.e.m). **F** Scatter plot of the relationship between response density (bouton fraction) and the mean response amplitude for each odorant separated by response polarity ($r = 0.81$ for suppressed boutons $P < 0.001$, chi-squared test; $r = 0.57$ for enhanced boutons, $P = 0.021$, chi-squared test). The underlying data for each plot are available in the source data file.

To visualize the selectivity of individual OSNs and boutons to each of the stimuli, we first rank-ordered the absolute value of responses to all 16 odorants and normalized them to the largest response (mean responses across all odorants, OSNs = 0.26 ± 0.06, boutons = 0.38 ± 0.07; Comparison of ranked distributions, $P < 0.001$; sign-rank test; Fig. 3B). Measurements of lifetime sparseness (see *Methods*) also indicated an increased turning breadth in cortical projections compared to OSNs (mean lifetime sparseness, 0.46 ± 0.004 in OSNs and 0.65 ± 0.002 in boutons; $P < 0.001$; Kolmogorov–Smirnov test; Fig. 3D). However, uniformly weak responses across a population could conflate relative population activities when normalized. To address this, we also measured population sparseness. Using this metric, we found that bouton responses were indeed denser than in OSNs (0.49 ± 0.01 boutons, 0.37 ± 0.04 OSNs; $P = 0.013$; sign-rank test; $n = 16$ odorants Fig. 3C, E), and these measurements were consistent across trials using the same odorants (Fig. 3G).

The responses of boutons to odorants were measured in awake mice, but OSN responses were acquired in anesthetized mice. To compare these two populations under similar conditions, we anesthetized animals with a cocktail of ketamine and xylazine and measured cortical feedback responses to the same panel of odorants (Supplementary Fig. 3). Under anesthetized conditions, feedback boutons were similarly tuned compared to awake animals, but the density of responses per odorant was reduced (Supplementary Fig. 3D–F). Despite the decreased density of response in anesthetized animals, we did not observe a systematic relationship between odorant tuning in anesthetized boutons and OSNs (Supplementary Fig. 3F, J).

We next estimated the representational similarity (see *Methods*) between pairs of odorants in our panel for OSNs and cortical boutons in the OB. In OSNs, a subset of the odorants showed similarity with other odorants in the panel, and relationships could be determined using hierarchical clustering of odorant-odorant correlations (Fig. 3H,

left). The odorant representations were well-preserved between trials of the same odorant (0.78 ± 0.03 mean correlation in OSNs; $n = 16$ odorants; Fig. 3I). However, in cortical projections, odorant responses in awake mice were more variable between trials (0.53 ± 0.02 mean correlation $P < 0.001$, sign-rank test; Fig. 3I), and the pairwise odorant relationships determined from OSNs did not map onto bouton pairwise odorant similarities (boutons to OSNs $r = -0.03$; $P = 0.59$; Fig. 3J). Interestingly, in anesthetized animals, we found a restructuring of odorant-odorant relationships and decreased trial variability (Supplemental Fig. 3H, J), yet there was no apparent relationship to representations in OSNs. The increased representational similarity between odorants in anesthetized animals could be due to reduced effective inhibition in the recurrent circuitry in the OB or the PC[55–57]. These data indicate cortical feedback axons are more broadly tuned to monomolecular odorants than OSNs, but have significantly decorrelated patterns of responses to different odorants than OSNs.

## Responses to complex odorant mixtures are normalized in the PC

The density of OSN activation can be systematically varied by delivering mixtures of monomolecular odorants with an increasing number of components[24]. We devised a panel of 84 mixtures derived from the 16 monomolecular odorants that were used earlier in our study (see Supplementary Table 3). The mixtures varied in complexity, contained 2, 4, 8, or 12 unique components, and were delivered to mice in pseudorandom order.

In OSNs, the activity and density of responses scaled, on average, with the number of odorant components within a mixture (Fig. 4Ai, Bi). However, despite a general increase in OSN activity with mixture complexity, the relationship is sublinear such that progressive increases in mixture complexity lead to increases in OSN activity at a diminishing rate[24,43,58]. We also identified a subpopulation of OSNs that

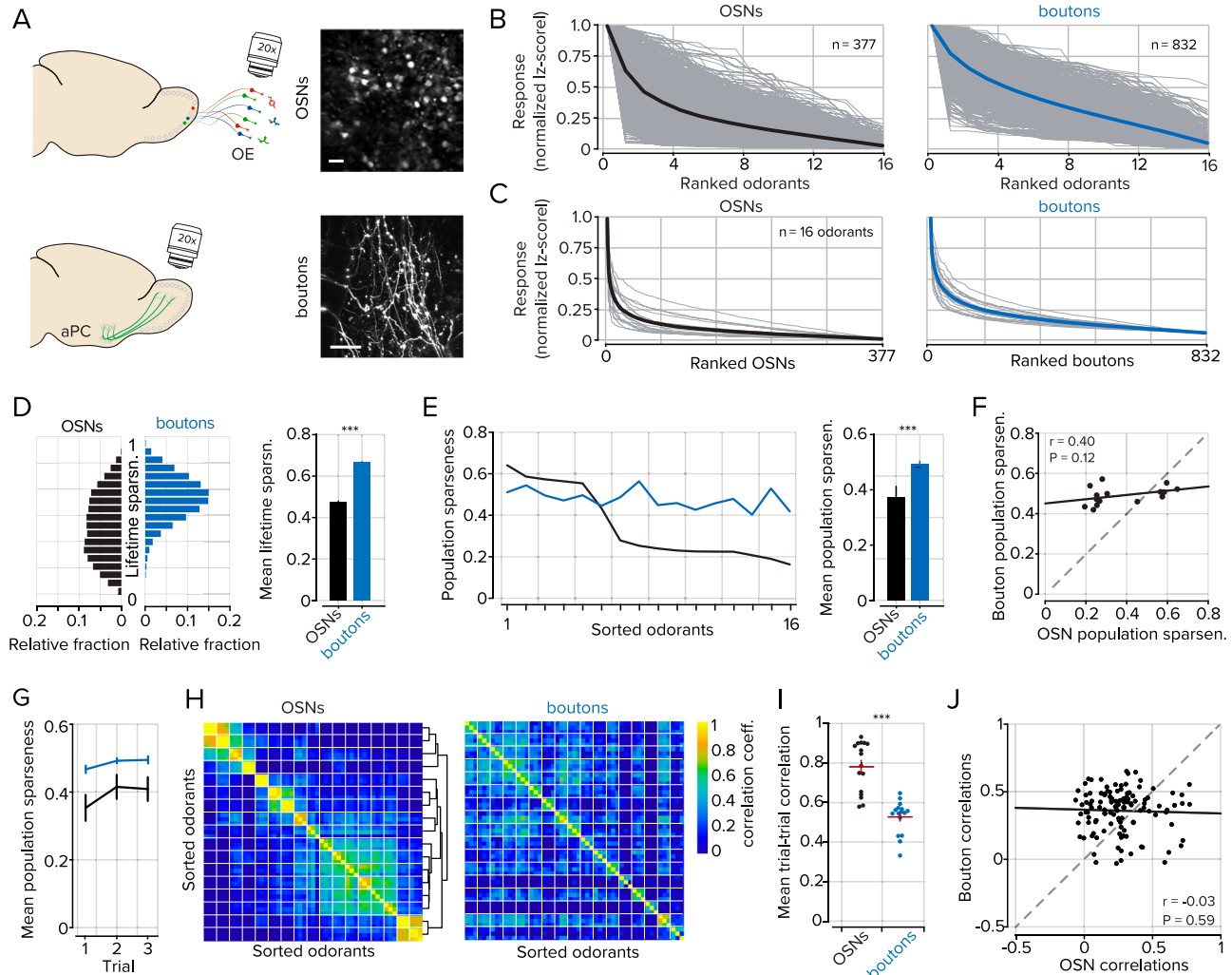

**Fig. 3 | Odorant representations in feedforward and feedback pathways to the OB. A** Top, Representative images of GCaMP3 expressing OSNs in the olfactory epithelium, scale bar = 20 μm. Bottom, GCaMP6f expressing cortical projections to the OB, scale bar = 20 μm. **B** Normalized and ranked responses to 16 odorants in OSNs (black; $n = 377$) and cortical projections to the OB (blue, $n = 832$). Each tuning curve is independently sorted and ranked. Gray lines represent individual ROIs and thick-colored lines represent the mean of all ROIs. **C** Normalized and ranked responses of each OSN and cortical bouton for each of the 16 odorants. Gray lines represent individual odorants, and thick-colored lines represent the mean of all odorants. **D** Left, Distributions of lifetime sparseness measured in OSNs (black, $n = 377$) and cortical projections (blue, $n = 832$). Right, mean lifetime sparseness was measured in OSNs and cortical projections. Kolmogorov–Smirnov test, error bars represent s.e.m. **E** Left, population sparseness for each of 16 odorants in OSNs (black) and boutons (blue) sorted to OSN values. Right, mean population sparseness for all odorants ($n = 16$). Sign-rank test, error bars represent s.e.m. **F** Scatter plot of the relationship between OSN population sparseness and bouton population sparseness. **G** Mean population sparseness for each of three trials ($n = 16$ odorants). Error bars represent s.e.m. **H** Odorant-odorant correlations in OSNs and boutons. Individual odorants are bounded by white lines, and each odorant contains three trials. Hierarchical clustering was used to group similar odorants in OSNs, and the clusters were then used to group datasets in boutons; see Supplemental Fig. 3 for odorant labels. **I** Variability within trials of the same odorants in OSNs and boutons. The horizontal red bar denotes the mean, and the vertical red bars represent s.e.m. ($n = 16$ odorants). **J** Scatter plot of the relationship between odorant-odorant correlations in OSNs and boutons. Chi-squared test. *** denotes $P < 0.001$. The underlying data for each plot are available in the source data file.

responded to odorant mixtures with decreases in activity[40] (Fig. 4Ai, right) and included these responses in our subsequent analyses when deviations from baseline activity met the inclusion criteria (see *Methods*).

We delivered the same panel of odorant mixtures to awake mice (three animals, five imaging fields), in the same order while imaging cortical projections to the OB. The mixture response distributions measured in the cortical feedback boutons had no relationship to the number of components in an odorant mixture and the mean activity of all activated boutons was similar at each mixture size (Fig. 4Aii-Bii). We next visualized the selectivity of individual OSNs and boutons to the range of mixture stimuli (Fig. 4C). For each element, we rank-ordered the absolute value of responses to all 100 stimuli and normalized them to the largest response. The rank-ordered bouton responses were

sparser than those of OSNs (Fig. 4C; comparison of mean distributions, $P < 0.001$; sign-rank test). The width of these "tuning curves" can also be characterized by measuring the lifetime sparseness (Fig. 4D). The lifetime sparseness of OSNs was strongly related to the mixture size – OSNs responded to more mixtures of a particular size as the size increased ($P < 0.001$, one-way ANOVA). While bouton lifetime sparseness did not scale with the number of components in a given mixture and was constant regardless of mixture size (Fig. 4D, right).

We next considered the population responses for each odorant mixture by rank-ordering them for each of the 100 mixture stimuli. While there was no difference between feedback boutons and OSNs in the ranked mean activity of all of the mixtures (Fig. 4E; Comparison of mean distributions, $P = 0.22$ Kolmogorov–Smirnov test), the distributions of individual odorant mixtures differed, with OSNs being more

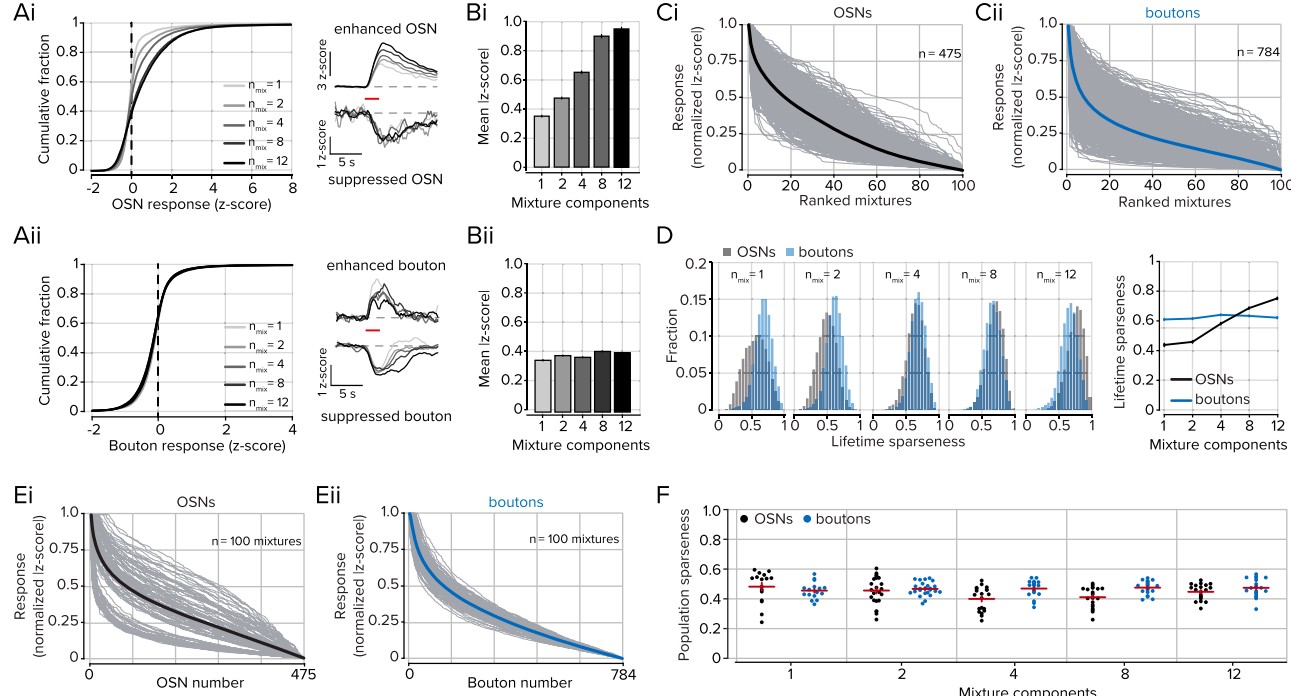

**Fig. 4 | Odorant mixtures are normalized by the PC. Ai-ii.** Left, distributions of OSN or bouton responses to odorant mixtures of increasing size. Grayscale shade corresponds to the mixture size. Right, for example, GCaMP responses of OSNs and boutons show enhanced or suppressed responses to odorant mixtures. Traces are averages of all significant responses of a given mixture size. Grayscale shade corresponds to the distributions on the left. The horizontal red bar indicates odorant delivery. **Bi-ii.** Mean activity at each mixture complexity in OSNs or boutons. **Ci-ii.** Normalized and ranked responses of 100 odorant mixtures in OSNs (black; $n = 475$) and cortical projections in awake mice (blue, $n = 784$). Each tuning curve is independently sorted and ranked. Gray lines represent individual ROIs, and thick-colored lines represent the mean of all ROIs. **D** Left, Distributions of lifetime sparseness for each group of mixtures of a given size ($n = 475$ OSNs, $n = 784$ boutons). Right, summary data of mean lifetime sparseness at each mixture size. Error bars represent s.e.m. **E** Normalized and ranked responses of each OSN and bouton for each of the 100 mixtures. Gray lines represent individual odorant mixtures and thick-colored lines represent the mean of all mixtures. **F** Population sparseness for each mixture of a given size ($n = 16$ individual odorants, 24 2-part mixtures, 20 4-part mixtures, 20 8-part mixtures, 20 12-part mixtures). The horizontal red bar denotes the mean and the vertical red bars represent s.e.m. The underlying data for each plot are available in the source data file.

variable than boutons. To estimate the spread of the mixture activity distributions, we measured its difference from the mean curve for each odorant mixture, averaging over the entire curve. Using this distance metric, we found that the population tuning for the 100 stimuli was significantly variable for OSNs, but was highly similar for boutons (Fig. 4E; OSN distance $= 0.11 \pm 0.01$, bouton distance $= 0.03 \pm 0.002$ in boutons; $P < 0.001$; Kolmogorov–Smirnov test). The stimulus tuning of each OSN or bouton can also be calculated by measuring population sparseness. For each of the mixture sizes, the mean population sparseness of OSNs and boutons was similar (Fig. 4F), yet when considering all mixtures OSNs had more variance in the sparsity of their responses ($P < 0.001$; $F$-test). Together, these data indicate that diverse stimuli that elicit highly divergent response sparsity in the OSNs are strongly equalized in cortical feedback axons.

**Representational similarity for mixture stimuli**

We next compared the representational similarity of odorant mixtures in feedforward and feedback inputs to the OB. In OSNs, the responses to mixtures became more similar as the number of odorants in a mixture increased (Fig. 5A), while in feedback projections, the representational similarity did not vary systematically with the mixture size (Fig. 5B). For each mixture, we then compared the relationship of pairwise mixture-mixture correlations between boutons and OSNs. Although there was a significant relationship between correlations found between OSNs and boutons, the slope of the regression was shallow and reflected the relatively narrower range of mixture-mixture correlations found in boutons ($r = 0.12$; $P < 0.001$; Fig. 5C).

To be certain that we did not introduce a sampling bias to our analysis by selecting an uneven number of boutons from different

axons, we also analyzed non-contiguous ROIs from each imaging field (See Methods). Our results did not change when we considered axon segments rather than boutons, and the representational similarity did not vary with the mixture size (Supplementary Fig. 4). Furthermore, boutons are small structures and comprise fewer pixels than OSNs. To address the possibility that the small number of pixels sampled from each bouton obscured relationships between odorant mixtures, we subsampled OSN ROIs to match boutons (Supplementary Fig. 5). Even when a single pixel was drawn from each OSN ROI, the structure in the mixture-mixture relationships remained, indicating the differences between OSNs and boutons are not due to the size of the analyzed regions.

For the two largest mixture sizes, we then compared the representational similarity in OSNs and feedback projections between highly overlapping mixtures, those that shared >=75% of their components, and other mixtures that had <75% overlap. In OSNs, mixtures that shared >=75% of their components had on average, more similar representations than mixtures that shared fewer components ($0.77 \pm 0.01$ mean correlation when mixture overlap >=75%, $n = 152$ mixture pairs $0.63 \pm 0.01$ mean correlation when mixture overlap <75, $n = 228$ mixture pairs; $P < 0.001$, Kolmogorov–Smirnov test; Fig. 5Di). In contrast, there was no difference in the mean correlation in cortical projections when mixtures contained greater or less than 75% overlap ($0.48 \pm 0.01$ mean correlation when mixture overlap >=75%, $0.46 \pm 0.01$ mean correlation when mixture overlap <75; $P = 0.06$, Kolmogorov–Smirnov test; Fig. 5Dii). These results indicate the representations of different mixtures become equally distinct from each other in cortical feedback, independent of mixture complexity.

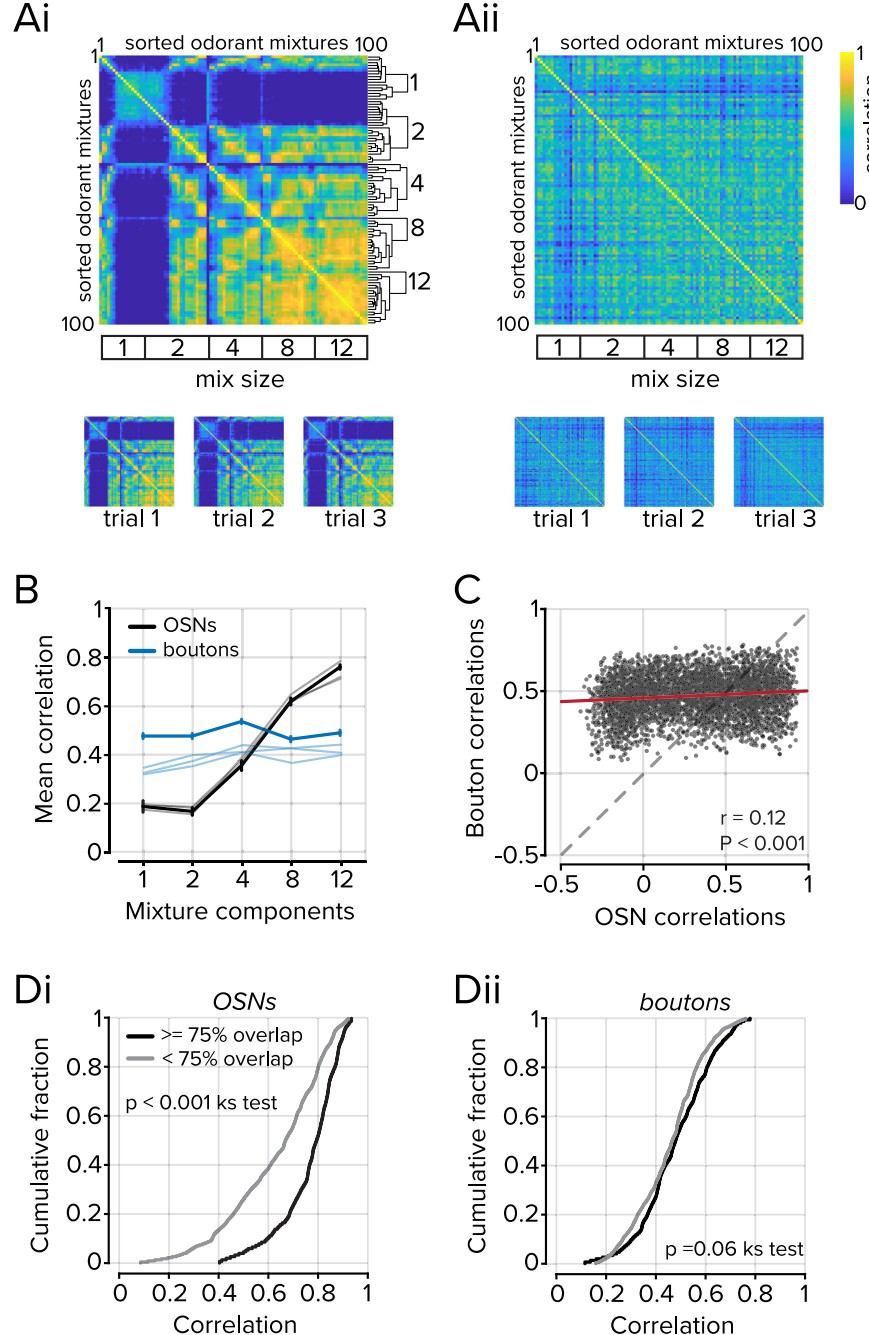

**Fig. 5 | Representations of odorant mixtures in OSNs and the PC. A** Correlation matrices of mixture-mixture relationships in OSNs (Ai) and boutons (Aii). Hierarchical clustering was used to group mixtures at each mixture size using OSNs. The clusters were then used to sort the bouton dataset. Bottom, Correlation matrices were obtained from three independent trials, showing similarity to the mean of all trials above. **B** Plot of the mean correlation of all mixtures of a given complexity ($n = 16$ individual odorants, 24 2-part mixtures, 20 4-part mixtures, 20 8-part mixtures, 20 12-part mixtures). Data from individual trials are plotted as shaded lines.

Error bars represent s.e.m. **C** Scatter plot of the relationship between mixture-mixture correlations in OSNs and boutons for each mixture size, chi-squared test. **D** Mixtures were divided into two groups based on mixture overlap using a threshold of 75%. Di. The OSN population activity was more correlated when mixtures shared >= 75% ($P < 0.001$, Kolmogorov–Smirnov test) Dii. In boutons, no difference was observed ($P = 0.06$, Kolmogorov–Smirnov test). The underlying data for each plot are available in the source data file.

## Nonmonotonic representations of odorant concentration in PC neurons

The activity of sensory cells at the periphery typically scales with stimulus intensity[59–61]. To corroborate this expectation, we imaged OSNs both at their somata in the OE and their axon terminals in the glomerular layer of the OB (Fig. 6Ai) in response to increasing odorant concentrations (see Supplementary Table 1, index 4 for odorant properties; Supplementary Fig. 6 for additional odorant). In OSNs,

population activity scaled with odorant concentration (Fig. 6Bi). Furthermore, as the odorant concentration increased, representations of the odorants became more similar at adjacent concentrations, approaching the similarity of responses between trials at the same concentration (Fig. 6Ci, D).

How do cortical projections to the OB respond to the same odorants at increasing concentrations? There is clear evidence of concentration invariance in neurons in the PC[36,37], but whether this

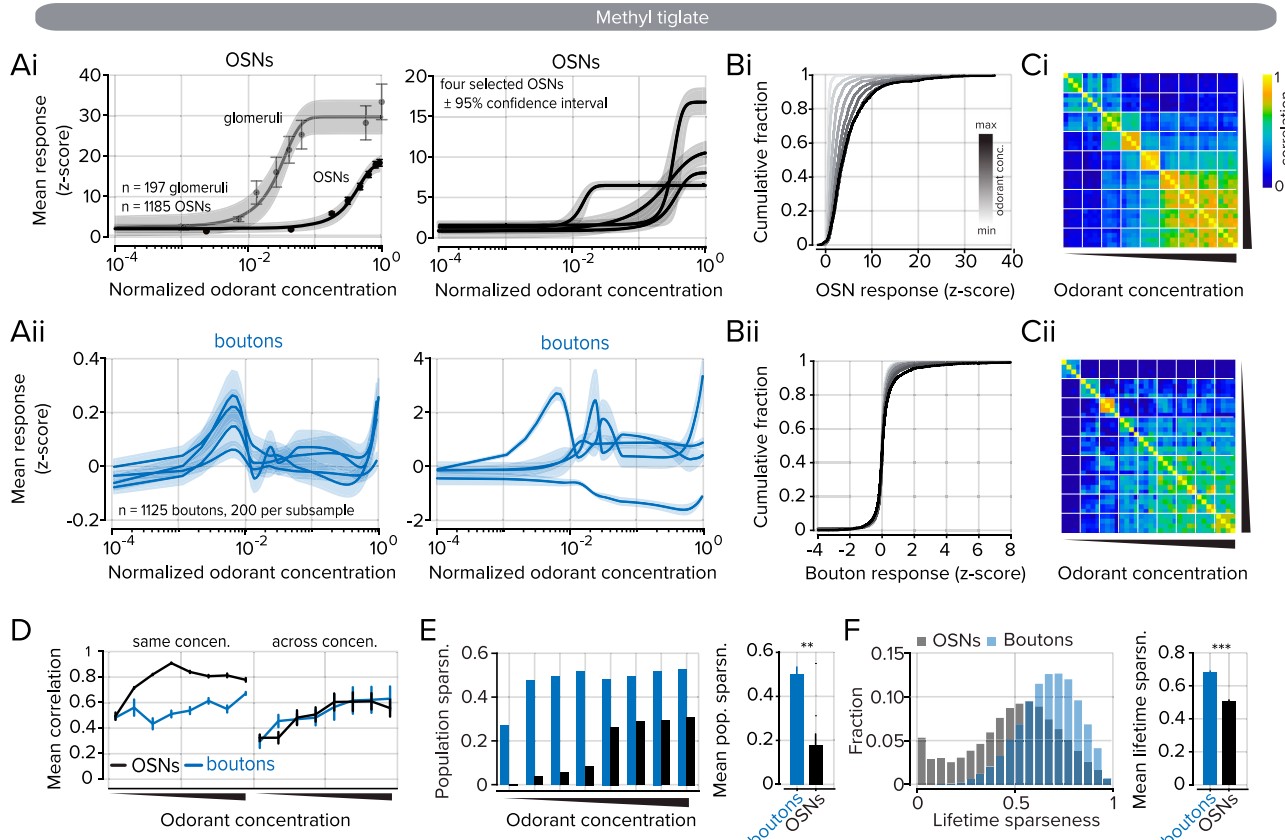

**Fig. 6 | Odorant responses in cortical projections to the OB are nonmonotonic.**
**Ai** Left, odorant concentration responses were measured in OSN somata in the OE and their axon terminals within the glomerular layer. Error expressed as s.e.m on individual data points. The solid line denotes a sigmoidal fit to the individual data points, and the shaded area denotes the 95% confidence interval for the sigmoidal fit. Right, four example OSN responses with the sigmoidal fit (solid) and 95% confidence interval (shaded) **Aii**. Left, odorant concentration responses were measured in cortical boutons in the OB. Four traces show subsamples selected from 200 boutons each. Data were fitted with an Akima piecewise cubic Hermite interpolation (solid line). The shaded area represents the 95% confidence interval. Right, four examples of individual bouton responses. **B** Distributions of OSN (Bi) and bouton (Bii) responses to increasing odorant concentration. Color shade corresponds to the odorant concentration. **C** Correlation matrix of odorant responses (each pixel is the pairwise correlation of the corresponding row and column elements) to increasing odorant concentration in OSNs (Ci) and boutons (Cii). White lines bound concentrations, and each concentration block contains four trials. **D** Left, OSN, and bouton correlations between trials at the same odorant concentration trials. Right, OSN and bouton correlations across concentrations ($n = 4$ trials). **E** Left, population sparseness at each odorant concentration for boutons (blue, $n = 8$ concentrations) and OSNs (black, $n = 8$ concentrations). Right, mean population sparseness for all concentrations. Sign-rank test, error bars represent s.e.m. **F** Left, Distributions of lifetime sparseness measured for all concentrations boutons (blue, $n = 1125$) and OSNs (black, $n = 1185$). Right, summary data of mean lifetime sparseness. Kolmogorov–Smirnov test, error bars represent s.e.m. ** denotes $P < 0.01$, *** denotes $P < 0.001$. The underlying data for each plot are available in the source data file.

invariance is reflected in back projections to the OB has not been investigated using a sufficiently large concentration range that is adequately sampled with intermediate points (some earlier studies used less than 2-fold changes in concentrations). We repeated the experiment, using the same odorant concentration range while imaging cortical feedback activity in the OB (Fig. 6Aii–Cii).

Bouton responses to increasing odorant concentrations could not be fitted with a sigmoidal function and lacked characteristic monotonic responses that were observed in OSNs (Fig. 6Ai vs. 6Aii). At the population level, these response properties are consistent with concentration-invariant odorant coding in the PC. However, many individual boutons displayed a clear preference for select and non-overlapping ranges of concentrations (Fig. 6Aii, right). Therefore, the information inherited from the PC in the OB may, on average, reflect concentration invariance, yet, at a more granular level, the PC may provide information on select concentrations. This observation substantiates other studies of odorant-concentration coding in feedback projections to the OB, which found that nonmonotonic concentration dependence was prevalent[12].

Like in the OSNs, the representational similarity of odorants measured in boutons increased as a function of concentration

(Fig. 6Cii). However, trial-to-trial variability within a concentration block was greater in boutons than in OSNs (Fig. 6D, left). In both OSNs and feedback boutons, we observed an increase in representational similarity, which scaled with odorant concentration and could reflect an increasing number of active cells (Fig. 6D, right). In the OSNs, the proportion of active cells, measured through population sparseness, had a strong relationship to odorant concentration and plateaued at the highest concentrations (Fig. 6E). However, the proportion of activated boutons did not scale with odorant concentration and was similar regardless of odorant concentration ($0.50 \pm 0.03$ mean population sparseness in boutons, $0.18 \pm 0.05$ mean population sparseness in OSNs; $n = 8$ concentrations; $P = 0.008$; sign-rank test). We then considered the tuning of each bouton or OSN to evenly distributed points throughout the concentration range to gauge their relative selectivity. On average, boutons were more widely tuned, as measured by lifetime sparseness, than were OSNs. While some OSNs displayed broad tuning, within the range of tuning seen for cortical boutons, another population was more selective and responded to only a single or few concentration points at the highest end of the concentration range (Fig. 6F; $0.69 \pm 0.01$ mean lifetime sparseness in boutons, $n = 1125$; $0.51 \pm 0.01$ mean lifetime sparseness in OSNs; $n = 1185$;

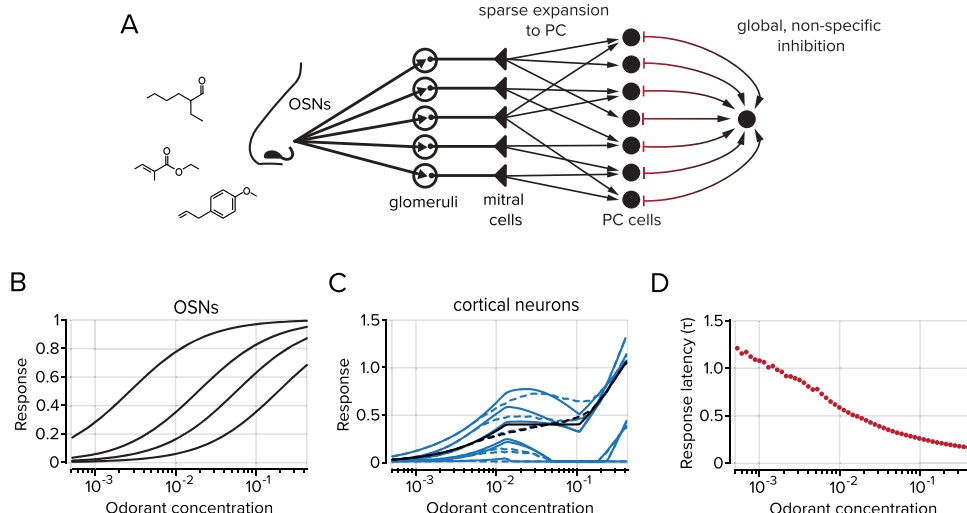

**Fig. 7 | A model of bulbo-cortial connectivity produces nonmonotonic concentration dependence in cortical neurons. A** Model schematic including sparse expansion from the OB to the PC and global, non-specific inhibition within the PC. **B** The response of olfactory sensory neurons is monotonic with odorant concentration, as empirically observed in Fig. 6. Each black line represents the activity of an individual OSN. **C** The activity of the feedback axons increases monotonically with concentration until global inhibition is activated, at which point axon activity decreases and then recovers. See Fig. 6Aii. Solid and dashed blue lines show dose-response curves for five representative cortical neurons in models with identical and heterogeneous inhibitory interneurons, respectively (see Methods for details on interneuron activity). Solid and dashed black lines show the average dose-response curves across all cortical neurons for these two models. **D** The response latency (in units of the membrane time constant) of cortical neurons decreases monotonically with concentration despite the nonmonotonic dependence on odorant concentration.

$P < 0.001$; Kolmogorov–Smirnov test). Our data indicate that cortical feedback axons bring complex, nonmonotonic information to the OB with increasing concentrations of individual odorants.

### Integrated activity-dependent inhibition explains non-monotonic concentration dependence in the PC

What accounts for the nonmonotonic concentration dependence observed in the cortical projections to the OB? To answer this question, we developed a minimal model of bulb-to-cortex circuit elements that incorporated global, activity-dependent inhibition within the PC[62] that can account for our experimental observations (Fig. 7A).

Specifically, we consider a sigmoidal dose-response model for the OSNs where an odorant binds to different OSN types with a broad range of affinities and, upon binding, activates a sparse subset of these OSNs (Fig. 7B). The response from olfactory sensory neurons is transformed linearly to the PC via the mitral cell population. This assumption about mitral cells is reasonable since published data indicate that mitral cell responses are gain-modulated by circuits in the OB but still exhibit monotonic responses to increasing concentrations of odor stimuli[27]. Our PC model consists of two populations of cells: one population that receives excitatory input from the OB and projects back to the OB and a second population of inhibitory interneurons that mediate global, non-specific inhibition. Inhibitory activity of the latter population is triggered when the summed activity of the former population exceeds a threshold. Below this threshold, the activity of the principal cells (and by extension, the feedback axons) increases monotonically with concentration until global inhibition kicks in, at which point their activity decreases (Fig. 7C). At sufficiently high concentrations, the excitatory input overcomes the inhibition, which leads to a second monotonic phase. Note that even though the dose-response curves of individual cells are nonmonotonic, the delay before inhibition acts implies that response latency decreases monotonically with concentration (Fig. 7D), as observed empirically[25].

## Discussion

The OB receives dense axonal input not only from the sensory periphery but also from cortical and associational areas of the brain. The stimulus tuning and response properties of peripheral input to the OB have been extensively described; however, less progress has been made on descending inputs to the OB and how they reflect the integrative and convergent architecture of the OB to PC circuitry. In our study, we systematically probed the tuning properties of cortical projections to the OB to understand how expanding sensory inputs are processed by the PC and represented in its descending inputs to the OB. We changed the density of OSN activity in two different ways—cumulative recruitment with increasing concentration and a more randomized increase with increased mixture complexity. We find that cortical axons bring back information to the OB that is highly equalized over a wide variety of stimuli, complementing the highly unbalanced input activity conveyed from the nose.

The PC is thought to create associative representations of the olfactory world and combine it with non-sensory information[45,63–66]. Circuits in the PC are likely to help normalize activity, creating equalized representations of diverse olfactory stimuli. Such normalization might also lead to concentration-invariant representations, at least when firing rate metrics are used[36]. It has not been clear what information computed in the PC is passed on to the OB via feedback axons. Previous studies using monomolecular odorants that have imaged cortical axons have noted that the signals are broad and spatially non-local[10,12]. Differences in activity in different brain states have also been reported, with larger odor-evoked responses in awake animals compared to anesthetized ones[10]. Intriguingly, task learning also appears to alter the activity patterns in cortical feedback[67–69]. Reducing the activity in cortical axons results in increased similarity of representations in mitral cells to different odors, suggesting that cortical axons ordinarily serve to decorrelate representations[12,70]. In this current study, we used a diverse set of odorants and created complex mixtures to mimic natural stimuli. By recording the responses of both OSNs and cortical feedback axons to the same set of diverse stimuli, we were able to directly compare their representations.

### Selectivity and variability of cortical feedback activity

We first characterized the odorant tuning of sensory neurons at the periphery and feedback projections to the OB, using a panel of

monomolecular odorants. Consistent with the circuit architecture expansion from the OB to the PC we found that cortical feedback projections were more broadly tuned to monomolecular odorants than sensory inputs to the OB. Interestingly, however, for complex mixture stimuli, cortical boutons were on average more selective than OSNs (see below). This arises because complex mixtures with many components activate OSNs densely (despite widespread antagonistic interactions), but the responses of cortical boutons have similar sparsity for a wide range of stimuli. This feature means that the choice of stimuli could be important for comparing the responses of OSNs and PC neurons or axons - monomolecular odorants, which are widely used in experiments, may not be representative of the wide range of response densities that natural mixtures may evoke.

A key feature of cortical responses we observed is the significant trial-to-trial variability. Our imaging procedures necessitated a smaller number of repeats than those used in electrophysiological studies, but our estimates of variability match those reported previously for cell bodies of PC neurons[25,37,71,72]. This higher variability was not just due to waking conditions, since cortical bouton responses in anesthetized mice were also more variable than OSN responses (Supplementary Fig. 3I). It remains unclear whether variability in stimulus encoding is a feature of cortical responsiveness to odorant stimuli, especially since this will be communicated back to the OB through descending projections. The zebrafish homolog of the olfactory cortex exhibits variability that results in changes to neuronal representations following each stimulus[73]. This variability is abolished by NMDA receptor antagonists, suggesting ongoing experience-dependent plasticity and drift in sensory cortical areas. A recent study, which compared data from mammalian and insect third-order neurons, has also proposed that stochasticity in responses may increase discriminability across odorant stimuli[72].

### Representational similarity of odors in cortical feedback axons

A widely proposed computational principle for many neural circuits and brain regions is that of pattern separation or pattern decorrelation[74–79]. In the olfactory system, this concept is applied in the context of decreasing the similarity of representations of distinct stimuli to allow for easier and more efficient decoding[75,79–81]. Odors activate overlapping sets of OSNs, and the similarity of their sensory representation is governed by the ligand-receptor binding properties, which depend partly on the physicochemical features of the ligands[23]. The dispersed, unstructured projections from the OB to the PC will decrease the similarity of stimulus representations, but theoretical analysis predicts that some similarity can be preserved even when projections are random[82,83]. That is, pairs of odors that are highly similar will have more similar representations in the PC than pairs of dissimilar odors. There is experimental evidence for this prediction, with a recent study reporting a relation between pairwise similarity in OB outputs and PC neurons[26]. However, it remains unclear how much pattern separation can still occur in the cortex, perhaps through experience and learning[84].

The relatively large number of unique stimuli generated in our study using mixtures afforded a wide range of representational similarities in the OSNs. Remarkably, the pairwise similarities observed in cortical boutons were only slightly related to the similarities in the inputs to the brain (in fact, there was no relation at all for monomolecular odorants). This finding suggests that representations in feedback axons are altered much more than what might be predicted by feedforward random projections from OB to PC[82]. For example, non-random associative connectivity within the PC[85] could decorrelate signals further, removing any remaining correlations predicted theoretically. In addition, cortical connectivity, either through experience or developmental biases, could build additional correlations absent in the OB representation or selectively attenuate certain correlations[26]. Our findings are also corroborated by recent work in insect brains, in which the similarity of representation in the output from the antennal lobe is not preserved in the mushroom body; instead, the representational similarity in the mushroom body seems to reflect covariances of odorant presence in natural odorant sources[86]. While our data point to overall decorrelation, they do not address whether specific correlations are selectively enriched or constructed in cortical feedback axons.

Our data also offer insights into the computations underlying the decorrelation of responses in the OB. Inhibiting feedback from the olfactory cortex has been shown to increase correlations in the representation of different monomolecular odors, suggesting that ordinarily, the activity of cortical feedback will serve to decorrelate representations[12,70]. Similarly, the activation of axons projecting from the raphe nucleus to the OB also decreases representational similarity[87]. The general inference from these studies, even if implicit, is that feedback information is global and distributed, and the selective recruitment of inhibition in the OB results in sparsening and decorrelation of M/T cell responses. In this circuit configuration, cortical inputs can shape granule cell activity such that a given M/T cell can be influenced by many more glomerular channels than its parent glomerulus alone. This, in turn, provides M/T cells access to global information about complex odorant environments.

Our data indicate that the feedback information is already significantly reformatted and decorrelated, which may reduce the demand for more specific circuitry in the OB. Related experimental work in zebrafish, and some theoretical ideas developed from it, suggest that decorrelation cannot be accomplished by global rescaling and instead requires more structured connectivity[80,88]. Cortical feedback axons carrying sparser decorrelated information, along with their plastic synapses[89,90], may facilitate pattern decorrelation in the OB.

### Olfactory cortical feedback axons carry normalized activity

Our understanding of sensory encoding in the visual system has benefited greatly from the use of natural stimuli. The advances include explanations of the shape of receptive fields in early visual areas as arising from the statistics of natural images[18] and sparse, decorrelated responses in the visual cortex elicited by natural images[20]. Similar experiments in the olfactory system have been rare, in part because of the difficulty in presenting natural stimuli in a controlled and reliable manner. In one step towards more naturalistic stimuli, we used diverse mixtures of commonly used odorants. Since these stimuli will span a range of covariances, it allowed us to test whether cortical representation has signatures of transformations expected from efficient coding.

Several features we observed support the idea of a more efficient representation of a mixture of stimuli in cortical feedback. First, population responses of cortical boutons were equalized for different stimuli. While the fraction of OSNs responding increased with mixture complexity, this fraction remained nearly constant in cortical boutons (Fig. 5D), significantly extending earlier work using monomolecular odorants or binary odorant mixtures and recording in the PC[25,35,37]. Recordings from anesthetized mice have indicated that responses of individual PC neurons to odorant mixtures can be described by a normalization model, where increasing input density gets progressively more attenuated[38]. In our studies, the activity generated by complex mixtures of odorants was massively normalized in cortical feedback axons arriving in the OB of awake animals. This is in stark contrast to OSNs, where the density of activity increases with the progressive complexity of stimulus mixtures encountered, even though this increase is highly nonlinear due to antagonistic interactions[24,43,91]. It is likely that the circuit architecture in the PC, with its feedforward and recurrent inhibition, serves to normalize and equalize responses.

A second signature of efficient coding is that the responses of individual cortical boutons were sparser than the responses of OSNs to

the panel of 84 mixtures. Interestingly, when responses to single odorants are compared, cortical boutons appear to be denser. This feature might be simply due to the particular choice of odorants and our ability to image only a small fraction of the entire OSN repertoire. Natural environments with mixtures of many odors are likely better approximated by our complex mixture stimuli, and those conditions will lead to the sparsening of representations in cortical feedback axons.

## Nonmonotonic representations of odorant concentration in the feedback axons

In contrast to the use of odorant mixtures, where antagonism contributes to the nonlinear scaling of activity in sensory inputs, increasing odorant concentrations provides a mechanism to scale the activity of OSNs independent of antagonistic interactions. As concentration increases, the same population of OSNs is increasingly activated and new OSNs are recruited[92] This contrasts with mixture stimulation, where more components can activate more OSNs, but without necessarily creating a gradual monotonic increase in the activity of OSNs.

A steady monotonic increase in the number and activity of OSNs as concentration increases might be predicted to result in increased activity of some cells in the PC. Previous studies have used the fraction of activated cells in the PC to argue for concentration-invariant normalization, with a slight dependence on concentration[36,37,93]. In our study, we see that individual cortical boutons show strong nonmonotonic dependence on odorant concentration (which cannot be construed as concentration invariant), even if the overall population response may be flatter. We also find that bouton representations of odorants increase in similarity with concentration. While this was hinted at in earlier studies[12,36], a systematic analysis has been lacking, as is a circuit-based explanation for such an observation.

We propose a simple cortical circuit model with activity-dependent global feedback inhibition that can explain the nonmonotonic dependence of cortical activity on concentration. This model is meant to be a plausible explanation, and including more biological realism in the future can allow more features of the data to be explained - for example, different concentrations at which distinct cortical neurons can exhibit maximal responses, as well as a more gradual decline in response amplitudes at mid-range of concentrations. We present our model to argue that known features of the olfactory circuitry can give rise to the seemingly paradoxical relationship between stimulus concentration and cortical feedback activity. A caveat in our interpretation is that earlier work has noted that M/T cells in the OB can also exhibit nonmonotonic concentration dependence[70]. However, this phenomenon is likely to be due to the influence of cortical feedback since inactivating it linearized M/T cell responses[70] (Fig. S8F). The effect is much stronger and more widespread in cortical boutons than in M/T cells, suggesting that this feature is not just simply inherited by PC neurons from the OB.

## Limitations of our study

Our study has some limitations. First, all the functional measurements reported are from calcium indicators, which can mainly track slow variations in activity and cannot easily reveal timing or latency measures faster than ~100 ms. However, a mitigating factor is that previous work has shown that average spike counts (or firing rate) carry much of the information in the PC[25,34]. For these reasons, we have reported our results by measuring changes in the calcium signal magnitudes rather than fine-scale differences in their temporal dynamics. A second limitation, resulting from the design of the study, is that we imaged a subpopulation of PC neurons, only those with feedback projections to the OB. Other principal neurons in the PC may have different properties. Previous work has shown that only deeper layer neurons send projections to the OB, and the superficial semilunar cells lack feedback

projections[94,95]. A recent study[71] indicated that semilunar and principal cells have many common properties, with only subtle differences in response tuning - therefore, we anticipate that the properties of PC neurons, extrapolated from bouton responses, may generalize to multiple types of principal cells. A third caveat is that the axonal and bouton activity could be influenced by local bulbar circuitry and may not faithfully represent somatic activity in PC neurons. For example, GABAb receptor-mediated presynaptic inhibition may suppress calcium responses locally[96]. Nevertheless, the net activity of cortical boutons, even if influenced by the bulbar environment, reflects the consequences for the postsynaptic targets within the OB and, therefore, functionally relevant. Similarly, nonlinear transformations between OSNs and M/T cells in the OB could result in some degree of equalization of sensory inputs prior to reaching the PC. Finally, in this study, we examined responses in anesthetized as well as awake mice, but with no behavioral outcomes required. It is possible that task learning and engagement change response properties since mice are likely to be in a more attentive state[68,69,97]. It is unlikely, however, that OSN responses are very different, except for being modified by sniff dynamics.

The interaction between bottom-up and top-down information streams in the olfactory system is likely to aid in interpreting complex sensory scenes. Naturalistic odor environments contain dozens or more unique odorants that must all be simultaneously parsed. Our studies demonstrate that cortical feedback maintains sparse odorant representations despite progressively dense sensory inputs, as likely to be encountered in natural environments. Whether sparse encoding, a hallmark of efficient neural processes, is maintained in cortical feedback as stimuli are assigned categorical relevance remains to be explored. Future studies using similar odorant delivery paradigms could explore how and if odorant representations in cortical projections are reformatted by learning and association.

## Methods

### Experimental model and subject details

Adult (>8 weeks) C57Bl/6 J or OMP-GCaMP3 (C57Bl/6 J background) mice of both sexes were used in this study. Sex was not considered in the study design. Mice were acquired from the Jackson Laboratory (C57Bl/6 J) or breeding stocks at Harvard University (OMP-GCaMP3) and maintained within Harvard University's Biological Research Infrastructure for the duration of the study. All animals were between 20 and 30 g before surgery and singly housed following any surgical procedure. Animals were between three and six months old at the time of the experiments. All mice used in this study were housed in an inverted 12-hour light cycle at 22 ± 1 °C at 30–70% humidity and fed *ad libitum*.

### Ethics oversight

All the experiments were performed in accordance with the guidelines set by the National Institutes of Health and approved by the Institutional Animal Care and Use Committee at Harvard University (protocol 29-20) or the University of Illinois Chicago (protocol 22-011).

### Viral injections

All viruses used in this study were acquired from Addgene. AAV9.-CAG.GCaMP6f.WPRE.SV40 (Addgene ID: 100836-AAV9) and AAV1-CAG-tdTomato (Addgene ID: 59462-AAV1). The two viruses were mixed in equal proportions prior to injection. Mice were anesthetized with an intraperitoneal injection of ketamine and xylazine (100 and 10 mg/kg, respectively) and the eyes were covered with petroleum jelly to keep them hydrated. Body temperature was maintained at 37 °C by a heating pad. The scalp was shaved and then opened with a scalpel blade. Two burr holes were then drilled above the anterior piriform cortex in each hemisphere. The coordinates for each of the injection sites are +1.2 or 1.6 mm AP relative to the intersection of the inferior

cerebral vein and superior sagittal sinus, +2.8 mm ML relative to the intersection of the inferior cerebral vein and superior sagittal sinus, and −3.6 or −3.2 mm DV from the brain surface. Viruses were infused at a rate of 40 nL/min for a total volume of 200 nL at each site from a 33-gauge beveled-tip needle (Hamilton). The scalp was then closed with dissolvable sutures. Buprenorphine SR-Lab (1.0 mg/kg) was administered subcutaneously, and the mice were allowed to recover for at least two weeks before any additional procedures.

## Confocal imaging

Mice were deeply anesthetized with a ketamine/xylazine mixture and transcardially perfused with 20 mL of PBS (pH 7.4) first, followed by 30-50 mL of 4% paraformaldehyde in 0.1 M phosphate-buffered saline (pH 7.4). Brains were removed and cut into 70 µm-thick sagittal sections using a vibratome (Leica). Slices were then washed and mounted for confocal imaging with DAPI mounting media and imaged with a confocal microscope (LSM 710 or 880, Zeiss).

## OB craniotomy

A craniotomy was performed to provide optical access to both OB. Mice were first anesthetized with an intraperitoneal injection of ketamine and xylazine (100 and 10 mg/kg, respectively), and the eyes were covered with petroleum jelly to keep them hydrated. Body temperature was maintained at 37 °C by a heating pad. The scalp was shaved and then opened with a scalpel blade. After thorough cleaning and drying, the cranial bones over the OBs were then removed using a 3 mm diameter biopsy punch (Integra Miltex). The surface of the brain was cleared of debris. The surface of the brain was kept moist with artificial cerebrospinal fluid containing in mM (125 NaCl, 5 KCl, 10 Glucose, 10 HEPES, 2 $CaCl_2$, and 2 $MgSO_4$ [pH 7.4]) and Gelfoam (Patterson Veterinary). Two 3 mm No. 1 glass coverslips (Warner) were glued together with optical adhesive (Norland Optical Adhesive 61) and adhered to the edges of the vacated cavity in the skull with Vetbond (3 M). The posterior portion of the exposed skull was gently scratched with a blade, and a titanium custom-made head plate was glued (Loctite 404 Quick Set Adhesive) on the scratches. C&B-Metabond dental cement (Parkell, Inc.) was used to cover the head plate and form a well around the cranial window. After surgery, mice were treated with carprofen (6 mg/kg) and buprenorphine SR-Lab (1.0 mg/kg). Animals were allowed to recover for at least three days prior to acclimatization in the imaging room.

## Bone thinning over the olfactory epithelium

OMP-GCaMP3 mice were anesthetized using the same procedure and all pre-surgical methods through head plate implantation are the same as the craniotomy. The cranial bones over the olfactory epithelium, anterior to the frontonasal suture, and between the internasal and nasal-maxillary sutures were thinned with a dental drill and scalpel blade until transparent[24,51]. The thinned area of the skull was then covered with cyanoacrylate adhesive (Loctite 404 Quick Set Adhesive) and a glass coverslip was implanted in the adhesive. Dental cement was then used to form a well over the thinned section of the skull. All animals were allowed to recover for at least three days before imaging experiments were initiated.

## Multiphoton imaging

A custom-built two-photon microscope was used for in vivo imaging. Fluorophores were excited and imaged with a water immersion objective (20X, 0.95 NA, Olympus) at 920 nm using a Ti:Sapphire laser with dispersion compensation (Mai Tai HP, Spectra-Physics). Images were acquired at 16-bit resolution and 4–8 frames/s. The pixel size was 0.6 µm for OSN somata and axon imaging. Fields of view ranged from 180 × 180 µm in the epithelium to 720 × 720 µm in the OB. The point-spread function of the microscope was measured to be 0.51 × 0.48 × 2.12 µm. Image acquisition and scanning were controlled by custom-written software in LabView (National Instruments). Emitted light was routed through two dichroic mirrors (680dcxr, Chroma, and FF555- DiO2, Semrock) and collected by a photomultiplier tube (R3896, Hamamatsu) using filters in the 500–550 nm range (FF01–525/50, Semrock).

## Odorant stimulation

Monomolecular odorants (Sigma or Penta Manufacturing) were used as stimuli and delivered by custom-built 16-channel olfactometers controlled by custom-written software in LabView[98,99]. For most experiments, the initial odorant concentration was 16% (v/v) in mineral oil, and further diluted 16 times with air. When using a concentration series, the initial odorant concentration was between 0.08%–80% (v/v) in mineral oil and further diluted 16 times with air and the relative concentration was measured by a photoionization detector (PID; Aurora Scientific), then normalized to the largest detected signal for each odorant (Supplementary Fig. 7). To create mixtures, air-phase dilution was used, and the total concentration of each odorant was held constant. For all experiments, the airflow to the animal was held constant at 100 mL/min, and odorants were injected into a carrier stream. Odorants were delivered 2–6 times each using a trial-based structure. In each trial, a five-second baseline period was followed by a two-second odorant delivery period. The intertrial interval between odorant deliveries ranged between 20–30 s.

For experiments characterizing the odor tuning, the odor panel consisted of (1) Ethyl tiglate (2) Allyl tiglate (3) Hexyl tiglate (4) Methyl tiglate (5) Isopropyl tiglate (6) Citronellyl tiglate (7) Benzyl tiglate (8) Phenylethyl tiglate (9) Ethyl propionate (10) 2-Ethyl hexanal (11) Propyl acetate (12) 4-Allyl anisole (13) Ethyl valerate (14) Citronellal (15) Isobutyl propionate (16) Allyl butyrate. See Supplementary Fig. 8 for PID measurements. For experiments measuring complex mixture responses in cortical projections and the olfactory epithelium, odorants 1–16 were used from the panel above. Additional odorant information is available in Supplementary Table 1, and the composition of odorant mixtures is found in Supplementary Table 3.

## Data analysis

Images were processed using both custom and available MATLAB (Mathworks) scripts. Motion artifact compensation and denoising were done using NoRMcorre[100]. The CaImAn CNMF pipeline Field[49] was used for bouton, epithelium, and axon imaging to select and demix ROIs. ROIs were further filtered by size and shape to remove merged cells. For signals obtained from glomeruli in the OB, custom scripts were written to manually select ROI boundaries[99]. The mean ΔF/F signal in the 5 s following odorant onset was used for measurements of neural activity in all experiments. To account for changes in respiration and anesthesia depth, correlated variability was corrected[58]. Thresholds for classifying responding ROIs were determined from a noise distribution of blank (no odorant) trials from which three standard deviations were used for responses. In each dataset, only ROIs with at least one significant odorant response were included for further analysis. Representational similarity between stimuli was estimated by calculating the Pearson correlation coefficient between population vectors that consisted of all ROIs that satisfied the thresholding criterion.

The expected fractions of boutons with only enhanced, suppressed, and mixed responses to all 16 odors were estimated from the overall response statistics obtained in Fig. 2B. For each simulated bouton, 16 responses were randomly and independently drawn from the observed probability distribution (72.2% non-responsive, 12.4% enhanced, and 15.4% suppressed) and were classified as purely enhanced, purely suppressed, or mixed. Expected values and variance from 10,000 such simulations were obtained.

Sparseness measures were calculated as reported by ref. 101. Population sparseness measures the fraction of elements (or cells) that

are activated by a given odorant, with values near one indicating uniform activity across all elements and values near zero indicating a lack of activity in most elements:

$$PS_j = \frac{\left(\sum_{i=1}^{n} \frac{r_{i,j}}{n}\right)^2}{\sum_{i=1}^{n} \frac{r_{i,j}^2}{n}} \tag{1}$$

Where: $n$ = the number of elements, $r_{i,j}$ = the response of a given OSN or bouton to odorant $j$.

Lifetime sparseness measures the extent to which a given element responds to different odorant stimuli. Values near one indicate all odorants uniformly activate a given element, and values near zero indicate a high degree of odorant selectivity:

$$LS_i = \frac{\left(\sum_{j=1}^{m} \frac{r_{i,j}}{m}\right)^2}{\sum_{j=1}^{m} \frac{r_{i,j}^2}{m}} \tag{2}$$

Where: $m$ = the number of odorants, $r_{i,j}$ = the response of a given OSN or bouton $i$ to odorant $j$.

All statistical comparisons for imaging experiments were made as described in the text for each figure and values are given as mean +/− standard error of the mean.

## Statistics and reproducibility
No statistical method was used to predetermine sample size, which was selected based on similar studies in the field (two-photon imaging). Identified regions of interest were excluded from the dataset if they were not significantly modulated (three standard deviations above a noise distribution) by any stimulus. The Investigators were not blinded to allocation during experiments and outcome assessment. For statistical analysis, normality tests were carried out before any statistical comparison. Appropriate parametric (or nonparametric) tests were chosen. If necessary, family-wise errors were corrected by dividing the initially chosen alpha (0.05) by the total number of comparisons. All statistical tests were performed as two-sided comparisons.

All imaging experiments consist of at least three trials per imaging field and each dataset consists of at least three independent imaging fields from three animals. All example images are from a single imaging field that provides an approximate representation of an entire dataset.

## OB-to-PC model
We consider a simplified model of connectivity between the OB and PC, which recapitulates the nonmonotonic dose-response curves shown by the cortical neurons. We model the activity of two neuronal populations: $N_g$ glomeruli in the OB and $N_p$ cortical neurons that project back to the OB.

The activity of the bulbar neurons reflects the binding and activation of OSN receptors. The output of the $i$th glomerulus to an odor at concentration $c$ is:

$$x_i(c) = \eta_i \frac{\kappa_i c}{1 + \kappa_i c}, \tag{3}$$

where $\kappa_i$ is the binding affinity of the odor to the receptors of the OSNs that project to the $i$th glomerulus and $\eta_i$ is proportional to its corresponding activation efficacy. The logarithms of the binding affinities, $\log \kappa_i$'s, were drawn independently and identically distributed (i.i.d) from a normal distribution with mean zero and standard deviation 3 so that the affinities spanned approximately three orders of magnitude. The activation efficacies, $\eta_i$'s, are binary and drawn i.i.d from a Bernoulli distribution with probability 0.2; that is, an odor activates approximately 20% of the glomeruli at saturating concentrations.

The dynamics of cortical activity are determined by bulbar input and global, non-specific inhibition through a population of local interneurons. Inhibition from the interneuron turns on when the summed activity of the cortical neurons exceeds a certain threshold. The voltage dynamics of the $j$th cortical neuron ($u_j$) after odor onset is given by:

$$\tau \frac{du_j}{dt} = -u_j - w_{inh}\sigma(\beta(v - v_{thr})) + \sum_{i=1}^{N_g} W_{ji}x_i, \tag{4}$$

where $v$ is the voltage of the inhibitory interneuron whose output activity is sigmoidal: $\sigma(\beta(v - v_{thr}))$. This neuron (or, equivalently, a population of identical neurons) non-specifically inhibits all the cortical neurons with synaptic weight $w_{inh}$. The bulbar input to the cortex is determined by the $N_p \times N_g$ sparse random matrix $W$ whose entries are non-zero with probability 0.1 and the non-zero entries are drawn from a positive half-normal distribution with scale 0.5. This latter number is set so that the input into the cortical neurons at saturating concentrations has unit magnitude on average. Note that changing this value does not affect the results if $w_{inh}$ is concomitantly scaled. A sparse $W$ ensures that the bulbar input across the cortical population has a broad distribution and thereby produces a distinct cortical representation for each odor. The output activity $y_j$ of the cortical neurons is rectified: $y_j = u_j^+$. $\tau$ is an integration timescale, which we expect to be on the order of a hundred milliseconds.

The inhibitory interneuron receives and sums input from all cortical neurons. The voltage dynamics of the inhibitory interneuron is:

$$\tau \frac{dv}{dt} = -v + \sum_{j=1}^{N_p} y_j \tag{5}$$

We set $w_{inh} = 1, \beta = 2, v_{thr} = 2000, N_g = 400, N_p = 5000$.

To simulate heterogeneous inhibition (Fig. 7C), we consider a population of 500 inhibitory interneurons in the PC. For each neuron, the three parameters, $w_{inh}$, $\beta$, and $v_{thr}$, were set to the values in the single neuron case above and were each scaled by a random factor of, $1 + \varepsilon$ where $\varepsilon$ is a normal random variable with mean zero and standard deviation 0.2.

## Reporting summary
Further information on research design is available in the Nature Portfolio Reporting Summary linked to this article.

# Data availability
The data supporting this study's findings have been deposited in a Github database (https://github.com/JDZak-Lab/Zak-et-al.−2024-Nat.-Comm.) under the https://doi.org/10.5281/zenodo.10697363[102]. Source data are provided in this paper.

# Code availability
supporting this study's findings have been deposited in a Github database (https://github.com/JDZak-Lab/Zak-et-al.−2024-Nat.-Comm.) under the https://doi.org/10.5281/zenodo.10697363.

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

## Acknowledgements
We thank Siddharth Jayakumar for his assistance with the confocal image collection, Ningjing Xia for thoughtful discussion on odorant mixtures, and all members of the Murthy and Zak Laboratories for helpful discussions. We also thank Bob Datta and Shyam Srinivasan for their helpful feedback on a draft of this manuscript. This work was supported by NIH Grants K99/R00 DC017754 to JDZ and R01 DC016289 to VNM.

## Author contributions
J.D.Z and V.N.M. Designed the experiments. J.D.Z. collected the data. J.D.Z., G.R., and V.K. analyzed the data. G.R. constructed the model. J.D.Z., G.R., and V.N.M. wrote and edited the manuscript with input from all authors.

## Competing interests
The authors declare no competing interests.
