## [Peer Review File · Nature Communications]

REVIEWER COMMENTS

Reviewer #1 (Remarks to the Author):

Zak et al address the interesting question of feedback processing in mammalian olfaction.

They do so by monitoring both OSNs and the projection boutons from the piriform onto granule cells in the olfactory bulb, and they deliver sixteen odors individually and in combinations, as well as at different concentrations. They observe that the dependency on several of these input parameters is different in OSNs and boutons, and they accordingly draw inferences on feedback processing. They finally make a simple model and relate it to their observations.

General comments:

The study is technically good and well designed to address the questions, but there are some limitations. First, the panel of odorants is limited to mostly esters. This leaves a small question mark over the generality of the findings. Possibly the authors could address this in the discussion. Second, the recordings are 2p, thus have a limited time-resolution, and a related issue is that it is lacking in behavioural context. The authors do discuss these points. Third, the conceptual and model framework could be stronger, as detailed below. Notwithstanding these caveats, the work has the potential to provide a significant advance in the field.

Major comments.

1. In a perfect world one would like to see not just the OSN responses, but also the M/T responses that go to the piriform. While I'm not asking for further experiments on this, it is a weak link in the authors' analysis of

feedback. Specifically, the olfactory signal going to the piriform is not the OSN activity, but the M/T cell activity. We don't know from the current study whether many of the observed differences between bouton and OSN responses may actually arise in the bulb itself. Given extensive published data on M/T activity, can the authors address this gap through discussion and ideally through modeling?

2. This point introduces another complication to the model. The feedback signal to the boutons modulates the M/T cell responses. This is missing in the model. I feel that the interpretation of the interesting concentration-dependent responses of boutons is weakened without this feedback. Note, for example, that the observed bouton responses do not necessarily have a second peak at high concentrations, which is what the model predicts. The width (concentration range) of the responses also is different from the model.

3. The authors almost entirely ignore dynamics of responses. From Figure 1 we see that there are interesting things happening in time on the bouton responses, such as some of them showing what looks like post-inhibitory rebound, and others having multiple phases of activity. We also know these features from the literature. Given that the authors have the data, can we see it? Can it inform their model and analysis somewhat more?

Specific comments.

1. Can the authors provide a clearer account of the timing of their protocol? From deep in the methods it looks like the odor pulse is 2 seconds (the bars in Fig 1 are too small to be sure) but it was not apparent to me what was the interval between odor presentations.

2. It would be nice to have a roughly-to-scale schematic of the two imaging preps, specifically the location of the cranial windows.

3. Over several figures we see several comparisons between OSN and bouton responses. It would be nice to collect these into a table.

Reviewer #2 (Remarks to the Author):

This is an interesting paper that uses Ca imaging to compare two different excitatory inputs to the olfactory bulb (feedforward and feedback) under two different odor stimulation paradigms (changing mixture complexity and changing concentration). The experiments appear to be carefully done and thoroughly analyzed. The main weakness is that, although the findings are nicely packaged, it is less clear if they are truly novel. The reader is left with the feeling that the broad conclusions are largely confirmatory, although the experiments do reveal some interesting details. Overall, however, this is an elegant piece of work that reinforces and clarifies our ideas about odor processing.

MAIN POINTS

1. Interpreting bouton fluorescence. Does bouton activity accurately reflect what is happening in the PC? Or is this irrelevant if all that matters is the nature of the feedback information that reaches the OB? It would be helpful if the authors make it clearer from the beginning where they stand on this distinction. If they do care that bouton activity faithfully represents somatic activity in PC neurons (see lines 555 ff), then more justification should be given at the start. For example, should the analysis treat boutons on the same and different axons differently in order to correctly infer PC activity? Given that the frame rate is only 4-8 Hz and the fast variant of GCaMP6 is used, how many spikes are missed and does it matter?

2. Novelty of the findings. If we accept that the bouton fluorescence gives an accurate readout of PC activity (see previous point), then the authors' findings indicate that PC activity is relatively insensitive to increasing mixture complexity and increasing odorant concentration. However, similar general conclusions have been made by other groups that have used electrical or imaging techniques to record directly from the PC (e.g. work from the groups of Wilson, Isaacson, Axel, Bekkers, Franks, Albeanu, etc.). Also, the authors' finding that OSN activity is graded seems unsurprising in the light of much prior work

on these neurons. That said, the authors do take a somewhat different approach here, and are certainly more systematic; however, giving more credit to previous work would not go amiss.

3. Functional importance of the two inputs. Although perhaps beyond the scope of this paper, it would be helpful to include some more discussion on how the two inputs interact in the bulbar circuit. At present this is only touched upon fleetingly (e.g. lines 503-504), but more comment on functional relevance would strengthen the significance of the findings.

MINOR POINTS

1. Lines 204 ff, variability between trials. What was the inter-trial interval? Could there have been variable cross-odor habituation?

2. Line 320, “using nearly identical odorant concentrations”. The meaning is unclear.

3. Lines 335 ff, increase in similarity between trials. This feature is unclear in Fig. 6D. Also lines 341-342, “tuning ... to the discrete points on the concentration axis”. This is unclear.

4. Discussion. Please refer to the relevant figure when discussing a finding.

5. Line 470 ff, raphe axons. Please clarify the meaning.

6. Lines 554-555, the results “are likely to generalize to other principal cells in the PC”. The logic of this statement is unclear.

7. Line 697. Clarify that r_i = the response of the i th element to odorant j . It’s unclear what A means here. See also line 703.

8. Line 721, an odor activates approx 20% of the glomeruli at saturating concentrations. How can this be, given the “one ORN class -> one or two glomeruli” rule, and the expectation that each odor activates a small number of ORN classes even at high concentration?

REVIEWER COMMENTS

Reviewer #1 (Remarks to the Author):

Zak et al address the interesting question of feedback processing in mammalian olfaction. They do so by monitoring both OSNs and the projection boutons from the piriform onto granule cells in the olfactory bulb, and they deliver sixteen odors individually and in combinations, as well as at different concentrations. They observe that the dependency on several of these input parameters is different in OSNs and boutons, and they accordingly draw inferences on feedback processing. They finally make a simple model and relate it to their observations.

General comments:

The study is technically good and well designed to address the questions, but there are some limitations. First, the panel of odorants is limited to mostly esters. This leaves a small question mark over the generality of the findings. Possibly the authors could address this in the discussion. Second, the recordings are 2p, thus have a limited time-resolution, and a related issue is that it is lacking in behavioural context. The authors do discuss these points. Third, the conceptual and model framework could be stronger, as detailed below. Notwithstanding these caveats, the work has the potential to provide a significant advance in the field.

We thank this reviewer for their overall positive remarks and agree that our manuscript will be strengthened by addressing these points.

Concerning esters and our odorant panel, the reviewer makes an important point. A major reason for the choice of esters is to activate the OSNs in the dorsal recess (and the dorsal OB) sufficiently to allow a robust study of population coding. We also note that there were two aldehydes and an anisole. There is little evidence that cortical responses are significantly different (in terms of overall statistics) for different classes of odorants. In fact, selecting our odorant panel to include 50% esters was a feature of our design. Our goal was to identify odorants that were represented similarly at the level of the OB and explore how those representations may be transformed by the piriform cortex. If we had selected 16 odorants that were distributed throughout chemical space, the probability of finding representationally similar odorants is greatly reduced.

In the context of the temporal resolution of microscopy and behavior, we agree with the reviewer about the limitations imposed by these choices we made for our study and will discuss them further. As for temporal resolution, we note that piriform responses can be well understood in terms of rate coding (Miura et al., 2012), with little evidence suggesting that temporal structure in responses carries substantial additional information. Nevertheless, it is a caveat we need to keep in mind. Behavioral context is important, and we are in the process of doing exactly these experiments for future studies (behaviors related to our previous work: Rokni et al., 2014 and Mathis et al., 2016).

Regarding our model, we understand that the reviewer feels that the model framework could be more comprehensive with the inclusion of M/T cell activity. We address this point below by providing limited empirical data from M/T cells demonstrating similar activity to OSNs when stimulated with odorant mixtures. In our opinion, our study is not undermined by the incomplete understanding of M/T cell activity since our goal was to demonstrate that distinct information is seen by the OB circuitry in the two major inputs – from the nose and the cortex. The reviewer provides an excellent rationale to motivate a follow-up study on how bulbar output neurons (M/T cells) integrate information from both streams, which we intend to pursue.

Major comments:

1. In a perfect world one would like to see not just the OSN responses, but also the M/T responses that go to the piriform. While I'm not asking for further experiments on this, it is a weak link in the authors' analysis of feedback. Specifically, the olfactory signal going to the piriform is not the OSN activity, but the M/T cell activity. We don't know from the current study whether many of the observed differences between bouton and OSN responses may actually arise in the bulb itself. Given extensive published data on M/T activity, can the authors address this gap through discussion and ideally through modeling?

This is an important point. We have done some experiments on M/T cells using similar sets of stimuli, but they are not extensive enough to be included in this study and the panel of odorant mixtures was not matched to the bouton and OSN datasets. Therefore, it is difficult to make a direct comparison between M/T cells and OSNs/boutons. Nevertheless, we believe that some general conclusions can be drawn. For example, from our experiments in M/T cells, we observe that their activity generally scales with mixture size (see Figure below). This observation is consistent with our data from OSNs and stands in contrast to the invariance we observe in cortical feedback projections. Therefore, we think it is reasonable to assume that, despite some degree of non-linear transformation from OSNs to M/T cells, for the most part, M/T cells reflect OSN activity.

A. Example GCaMP6f resting fluorescence imaged in M/T cells. Scale bar = 50 μm . **B.** Distributions of M/T cell responses to odorant mixtures of increasing size. Grayscale shade corresponds to the mixture size. **C.** Mean activity at each mixture complexity in M/T cells.

We agree that these should be discussed since it is important to understand how the activity of OSNs is transformed in the OB before the information is passed on to the cortex. At the same time, we feel that it will be very difficult to sufficiently constrain any model of the OB to predict the responses of M/T cells to the complex stimuli we have used. There are many nonlinear steps involved in the cellular and synaptic properties within the OB itself, not to mention the complex and poorly understood wiring diagram within this region. We prefer to acknowledge this gap in our paper and leave it to future studies to report the M/T cell activity to similar stimuli. We have included the following text to our discussion at line 592, “Similarly, nonlinear transformations between OSNs and M/T cells in the OB could result in some degree of equalization of sensory inputs prior to reaching the PC.”

On a conceptual level, we feel that the key point of our paper is the demonstration that distinct information is seen by the OB circuitry in the two major inputs. That demonstration is not, in our opinion, undermined by the incomplete understanding of M/T cell activity in this context (even though that would be very helpful indeed).

2. This point introduces another complication to the model. The feedback signal to the boutons modulates the M/T cell responses. This is missing in the model. I feel that the interpretation of the interesting concentration-dependent responses of boutons is weakened without this feedback. Note, for example, that the observed bouton responses do not necessarily have a second peak at high concentrations, which is what the model predicts. The width (concentration range) of the responses also is different from the model.

We acknowledge that our model does not predict the granular details of bouton activity to different concentrations, such as the width of the responses. Given the simplicity of our model, it would be quite surprising if we got all the details correct. Our model is presented mainly to argue that relatively straightforward features of the known circuitry can give rise to the seemingly paradoxical (general) relationship. The one study that examined the concentration dependence of M/T cell activity using very similar methods (Banerjee et al., 2015), found that M/T cell activity has largely monotonic responses to increasing concentration of odors (their Figure 7CD; some responses could be declining, but still are monotonic). We acknowledge, of course, that the responses to increasing odorant concentration in M/T cells will be altered from those of OSNs. But our key point is that the particularly paradoxical feature of non-monotonic responses in cortical boutons is unlikely to be explained by M/T cell responses. As for the second peak, we call attention to the examples shown in Figure 6Aii, which clearly show boutons that have a rise in response amplitude at the higher concentrations. In fact, this observation is relatively common and is one of the interesting features that prompted us to consider a model to explain the phenomenon.

3. The authors almost entirely ignore dynamics of responses. From Figure 1 we see that there are interesting things happening in time on the bouton responses, such as some of them showing what looks like post-inhibitory rebound, and others having multiple phases of activity. We also know these features from the literature. Given that the authors have the data, can we see it? Can it inform their model and analysis somewhat more?

We thank the reviewer for calling attention to this. Indeed, we too noticed this striking feature. However, the reviewer themselves brings up a crucial issue above in the “General Comments” section – the slow temporal resolution of Ca imaging. Given this limitation, we have chosen not to focus our attention on this aspect. I hope the reviewer agrees with this assessment, and we are glad to add a note to this effect in the Discussion. At line 579 we include the revised caveat, “For these reasons, we have reported our results by measuring changes in the calcium signal magnitudes rather than fine-scale differences in their temporal dynamics.”

Specific comments:

1. Can the authors provide a clearer account of the timing of their protocol? From deep in the methods it looks like the odor pulse is 2 seconds (the bars in Fig 1 are too small to be sure) but it was not apparent to me what was the interval between odor presentations.

Apologies for not making this clear – we have now noted this in key places.

Under the heading, “*Odorant tuning profiles of cortical projections to the OB,*” at line 117, we have modified the following text, “Odorants were delivered for two seconds each in a pseudorandom order with intertrial intervals between odorants of at least 20 seconds.”

In the methods under the “*Odorant Stimulation*” subheading, we have added the following text: “Odorants were delivered 2–6 times each using a trial-based structure. In each trial, a five-second baseline period was followed by a two-second odorant delivery period. The intertrial interval between odorant deliveries ranged between 20–30 seconds.”

2. It would be nice to have a roughly-to-scale schematic of the two imaging preps, specifically the location of the cranial windows.

We thank the reviewer for this suggestion. We now include a new supplemental figure that provides a schematic of the imaging window locations. The figure is referenced in the text related to main Figure 3 and included here for convenience.

3. Over several figures we see several comparisons between OSN and bouton responses. It would be nice to collect these into a table.

This is a helpful organizational suggestion from this reviewer. In our revised manuscript we now provide tables related to several metrics that are compared between OSNs and cortical feedback boutons. Please see the new tables that are associated with Figures 3, 4, and 6.

Reviewer #2 (Remarks to the Author):

This is an interesting paper that uses Ca imaging to compare two different excitatory inputs to the olfactory bulb (feedforward and feedback) under two different odor stimulation paradigms (changing mixture complexity and changing concentration). The experiments appear to be carefully done and thoroughly analyzed. The main weakness is that, although the findings are nicely packaged, it is less clear if they are truly novel. The reader is left with the feeling that the broad conclusions are largely confirmatory, although the experiments do reveal some interesting details. Overall, however, this is an elegant piece of work that reinforces and clarifies our ideas about odor processing.

We appreciate this reviewer's enthusiasm for our experimental and analytical work, as well as its role in clarifying ideas about odor processing. However, we contend that our work extends beyond simple confirmation of previous studies. In one respect, we do agree with the reviewer that a subset of our work confirms past studies of concentration invariance in the piriform cortex (for example, Bolding and Franks eLife 2017; Roland et al eLife 2017); however, providing such confirmation is necessary to support our observations on the more granular details of the relationship between cortical activity and odorant concentrations. For example, non-monotonic concentration preferences in boutons. Furthermore, to our knowledge, no studies have explored how the piriform cortex encodes mixtures of varying complexity.

We appreciate the constructive feedback from the reviewer in the following comments and we feel that addressing these comments has strengthened our manuscript.

MAIN POINTS

1. Interpreting bouton fluorescence. Does bouton activity accurately reflect what is happening in the PC? Or is this irrelevant if all that matters is the nature of the feedback information that reaches the OB? It would be helpful if the authors make it clearer from the beginning where they stand on this distinction. If they do care that bouton activity faithfully represents somatic activity in PC neurons (see lines 555 ff), then more justification should be given at the start. For example, should the analysis treat boutons on the same and different axons differently in order to correctly infer PC activity? Given that the frame rate is only 4-8 Hz and the fast variant of GCaMP6 is used, how many spikes are missed and does it matter?

The reviewer brings up an important point with the speed of imaging and the decay time of GCaMP6f. However, given the shape of the responses we observe, we are unlikely to miss a significant number of spikes. Unlike in brain regions such as the hippocampus, responses in the PC are much more graded and not temporally sparse. We performed some simulations to estimate how PC spike responses obtained from the literature (including our own work) would appear filtered by GCaMP6f kinetics and our imaging sampling rate. In the figure below, we simulate GCaMP6f signals when sampled at a 5 Hz frame rate. We used Poisson distributed spike trains at 10 Hz baseline, which rise abruptly to 30 Hz to mimic odor stim (*top*). The spikes are then run through a convolution with 150ms decay (the reported GCaMP6f decay time constant; *middle*). The convolved signal is sampled at 5 Hz with 100 different phase shifts (*bottom*). In each of the phase shifts, a clear increase in the sample signal which is locked with the simulated odorant delivery. The reviewer is correct that some spikes may have been missed by our sampling rate, but with these simulations, we contend that the effect of missed spikes is negligible. We explicitly noted the temporal filtering by calcium indicators in the manuscript in the "Limitations of our study" section.

2. Novelty of the findings. If we accept that the bouton fluorescence gives an accurate readout of PC activity (see previous point), then the authors' findings indicate that PC activity is relatively insensitive to increasing mixture complexity and increasing odorant concentration. However, similar general conclusions have been made by other groups that have used electrical or imaging techniques to record directly from the PC (e.g. work from the groups of Wilson, Isaacson, Axel, Bekkers, Franks, Albeanu, etc.). Also, the authors' finding that OSN activity is graded seems unsurprising in the light of much prior work on these neurons. That said, the authors do take a somewhat different approach here, and are certainly more systematic; however, giving more credit to previous work would not go amiss.

We respectfully disagree with the reviewer. All the work cited by the reviewer has used binary mixtures at best and to our knowledge the only work that has reported activity in the PC to larger mixtures is Penker et al., 2020 and that was in anesthetized animals. Therefore, we are not sure to which work the reviewer is referring. Perhaps the reviewer feels it is reasonable to automatically extend what has previously been observed for binary mixtures and narrow concentration ranges to complex stimuli and a wide range of concentration – we argue that it is better to empirically confirm this. Our sense is vindicated by our finding of non-monotonic dependence of bouton responses to increasing concentration, which was certainly not reported by previous work. We are glad to attribute appropriate credit to previous pioneering work.

3. Functional importance of the two inputs. Although perhaps beyond the scope of this paper, it would be helpful to include some more discussion on how the two inputs interact in the bulbar circuit. At present this is only touched upon fleetingly (e.g. lines 503-504), but more comment on functional relevance would strengthen the significance of the findings.

We agree with the reviewer's initial assessment that a discussion of the interactions between the two inputs at the level of the OB is beyond the scope of this work. While we do indeed agree that many interesting experiments could address this question, any discussion of how the two inputs affect M/T cell responses would be overly speculative without any accompanying empirical evidence. To address this comment on functional relevance, we have added the following text to our discussion at line 501, "In this circuit configuration, cortical inputs can shape granule cell activity such that a given M/T cell can be influenced by many more glomerular channels than its parent glomerulus alone. This, in turn, provides M/T cells access to global information about complex odorant environments."

MINOR POINTS

1. Lines 204 ff, variability between trials. What was the inter-trial interval? Could there have been variable cross-odor habituation?

We apologize for not including these details in the Methods – we have done so now. The inter-trial interval was between 20-30 sec. Please see "specific comment 1" from the other reviewer for the modified text. We do not think there was systematic cross-odor habituation since the presentations were effectively randomized. However, if there were some very specific interactions between some particular pairs of odors, it would be hard to detect

in our protocol. We note that the variability was much higher in bouton responses, and the OSN responses were much more reliable across trials arguing against peripheral habituation. We also note that the first response was consistently denser than the other two (Figure 3G), suggesting that there may have been a general effect of familiarity.

In our discussion, we now point to recent work that observes variability in the zebrafish homolog of the olfactory cortex. At line 454, the following text is included, “The zebrafish homolog of the olfactory cortex exhibits variability that results in changes to neuronal representations following each stimulus (Jacobson et al., 2018). This variability is abolished by NMDA receptor antagonists, suggesting ongoing experience-dependent plasticity and drift in sensory cortical areas.”

2. Line 320, “using nearly identical odorant concentrations”. The meaning is unclear.

We intended to indicate that the concentration range and points sampled were closely aligned with the OSN data. We agree with the reviewer that the statement is unclear and have revised the text to say, “We repeated the experiment, using the same odorant concentration range while imaging cortical feedback activity in the OB”

3. Lines 335 ff, increase in similarity between trials. This feature is unclear in Fig. 6D. Also lines 341-342, “tuning ... to the discrete points on the concentration axis”. This is unclear.

We have added the following text at lines 354 and 361 to address our previous ambiguity.

“In both OSNs and feedback boutons, we observed an increase in representational similarity which scaled with odorant concentration and could reflect an increasing number of active cells (**Figure 6D, right**).”

“We then considered the tuning of each bouton or OSN to evenly distributed points throughout the concentration range to gauge their relative selectivity.”

4. Discussion. Please refer to the relevant figure when discussing a finding.

We have done so as much as possible, without being distracting and redundant. Each figure is called out in the order of its panels within the results section. At several points in the discussion we refer to specific figures, but referencing each of the seven main figures is not practical.

5. Line 470 ff, raphe axons. Please clarify the meaning.

We have expanded this to clarify: “activation of axons projecting from the raphe nucleus to the OB...”

6. Lines 554-555, the results “are likely to generalize to other principal cells in the PC”. The logic of this statement is unclear.

We have modified this to be clearer: “... differences in response tuning - therefore, we anticipate that the properties of PC neurons extrapolated from bouton responses may generalize to multiple types of principal cells.”

7. Line 697. Clarify that r_i = the response of the i th element to odorant j . It's unclear what A means here. See also line 703.

We thank the reviewer for pointing this out. In this case “ A ” refers to a given OSN or bouton from our datasets, which should have been subscripted with i . Therefore we have modified the text to say, “ r_{ij} = the response of a given OSN or bouton i to odorant j ”

8. Line 721, an odor activates approx 20% of the glomeruli at saturating concentrations. How can this be, given the “one ORN class -> one or two glomeruli” rule, and the expectation that each odor activates a small number of ORN classes even at high concentration?

We are not sure what the reviewer means by this comment. The number of OSN subtypes that are activated by a single odorant is highly dependent on concentration, as well as other factors. There are several papers showing that a large fraction of OSNs or glomeruli are activated by odors at high concentrations (eg. Vincis et al., 2012, Wachowiak and Cohen 2001, Meister and Bonhoeffer, 2001). Vincis and Colleagues report a 5x increase in the number of active glomeruli over odorant dilutions ranging from 1/100 to 1/20 (their Supplementary Figure 9B). Meister and Bonhoeffer make a similar observation and remark on the vast increase in the number of activated glomeruli to odorant concentration with the following passage, “To distinguish odors C6, C7, and C8 at high concentration ($D = 0.1$), it is clearly not sufficient to find the glomerulus with the strongest response. Instead it becomes more revealing to identify which glomeruli in the set are not activated by the odor.”

REVIEWERS' COMMENTS

Reviewer #1 (Remarks to the Author):

The authors have addressed most of my comments well. I have one minor change to recommend: Comment 2 (about feedback from PC to M/T cells) is discussed in the response but these points do not seem to make it into the MS. It would be useful for the reader to have this reasoning included in the text, specially since there is already a section or two about PC feedback into the OB (Lines 494 to 510, also lines 569-572).

Reviewer #2 (Remarks to the Author):

I thank the authors for their careful consideration of my feedback. The paper is definitely improved.

Regarding my earlier comments about novelty, I apologize for being vague. On odor mixture size, the work by Penker et al., 2020, that the authors mention has looked at the effect of larger mixtures in the PC, and also earlier work by Barnes et al. (2008) and Chapuis & Wilson (2012), which used 10-component mixtures. It's true that these experiments used anesthetized animals. If this is a key distinction, then it should be spelled out. On the effect of concentration changes in the PC, there are a number of previous studies that used wide ranges of concentration, e.g. Stettler & Axel, 2009 (25x), Bolding & Franks, 2018 (33x), Tantirigama et al., 2017 (100x), Otazu et al, 2015 (1000x).

As I said before, the authors' work goes beyond all of these, and their work is elegant and important. It's really up to them to decide on the right amount of credit to give to earlier studies.

Reviewer #1 (Remarks to the Author):

The authors have addressed most of my comments well. I have one minor change to recommend: Comment 2 (about feedback from PC to M/T cells) is discussed in the response but these points do not seem to make it into the MS. It would be useful for the reader to have this reasoning included in the text, specially since there is already a section or two about PC feedback into the OB (Lines 494 to 510, also lines 569-572).

We appreciate the reviewer's feedback and enthusiasm for our manuscript. We are unsure of which text the reviewer is referring to in this comment. In our previous revision, we directly incorporated our response to their Comment 2 at (previous) lines 569-572. Nevertheless, we have added additional text to the section titled "*Nonmonotonic representations of odorant concentration in the feedback axons*" to further elaborate our point about M/T cells inheriting non-monotonic concentration dependence from the PC.

"We propose a simple cortical circuit model with activity-dependent global feedback inhibition that can explain the nonmonotonic dependence of cortical activity on concentration. This model is meant to be a plausible explanation, and including more biological realism in the future can allow more features of the data to be explained - for example, different concentrations at which distinct cortical neurons can exhibit maximal responses, as well as a more gradual decline in response amplitudes at mid-range of concentrations. We present our model to argue that known features of the olfactory circuitry can give rise to the seemingly paradoxical relationship between stimulus concentration and cortical feedback activity."

Reviewer #2 (Remarks to the Author):

I thank the authors for their careful consideration of my feedback. The paper is definitely improved.

Regarding my earlier comments about novelty, I apologize for being vague. On odor mixture size, the work by Penker et al., 2020, that the authors mention has looked at the effect of larger mixtures in the PC, and also earlier work by Barnes et al. (2008) and Chapuis & Wilson (2012), which used 10-component mixtures. It's true that these experiments used anesthetized animals. If this is a key distinction, then it should be spelled out. On the effect of concentration changes in the PC, there are a number of previous studies that used wide ranges of concentration, e.g. Stettler & Axel, 2009 (25x), Bolding & Franks, 2018 (33x), Tantirigama et al., 2017 (100x), Otazu et al, 2015 (1000x).

As I said before, the authors' work goes beyond all of these, and their work is elegant and important. It's really up to them to decide on the right amount of credit to give to earlier studies.

We thank the reviewer for their positive feedback. In light of these comments, we have included new text to our discussion that emphasizes that our study was carried out using awake animals. We have cited each of the studies suggested by this reviewer.